



# New particle formation in coastal New Zealand with a focus on open ocean air masses

Maija Peltola[1], Clémence Rose[1], Jonathan V. Trueblood[1], Sally Gray[2], Mike Harvey[2], and
Karine Sellegri[1]

[1]Laboratoire de Météorologie Physique (LaMP-UMR 6016, CNRS, Université Clermont Auvergne), 63178, Aubière, France
[2]National Institute of Water & Atmospheric Research Ltd (NIWA) Private Bag 14-901, Wellington, New Zealand

**Correspondence:** Maija Peltola (m.peltola@opgc.fr) and Karine Sellegri (k.sellegri@opgc.univ-bpclermont.fr)

**Abstract.**

Even though oceans cover the majority of the Earth, most aerosol measurements are from continental sites. We measured aerosol particle number size distribution at Baring Head, in coastal New Zealand, over a total period of 10 months to study aerosol properties and new particle formation, with a special focus on aerosol formation in open ocean air masses. Particle

concentrations were higher in land-influenced air compared to clean marine air in all size classes from sub-10 nm to cloud condensation nuclei sizes. When classifying the particle number size distributions with traditional methods designed for continental sites, new particle formation was observed at the station throughout the year with an average event frequency of 23%. While most of these traditional event days had some land-influence, we also observed particle growth starting from nucleation mode during 16 % of the data in clean marine air and at least part of this growth was connected to nucleation in the marine

boundary layer. Sub-10 nm particles accounted for 29% of the total aerosol number concentration of particles larger than 1 nm in marine air during the spring. This shows that nucleation in marine air is frequent enough to influence the total particle concentration. Particle formation in land-influenced air was more intense and had on average higher growth rates than what was found for marine air. Particle formation and primary emissions increased particle number concentrations as a function of time spent over land during the first 1-2 days spent over land. After this, nucleation seems to start getting suppressed by the

pre-existing particle population, but accumulation mode particle concentration keeps increasing, likely due to primary particle emissions. Further work showed that traditional NPF events were favoured by sunny conditions with low relative humidity and wind speeds. In marine air, formation of sub-10 nm particles was favoured by low temperatures, relative humidity, and wind speeds and could happen even during the night. Our future work will study the mechanisms responsible for particle formation at Baring Head with a focus on different chemical precursor species. This study sheds light on both new particle formation in

open ocean air masses coming from the Southern Ocean and local aerosol properties in New Zealand.

## 1 Introduction

Marine aerosols are a core component of the global climate system. The oceans cover over 70 % of the Earth and can be considered as a relatively dark surface. Aerosol concentrations and properties largely determine how much of the oceans is





covered by haze or clouds and how bright the clouds are. A recent study showed that aerosols can explain 75 % of low-level
marine clouds' cooling effects (Rosenfeld et al., 2019). While primary sea spray aerosols are important and at high wind speeds
they can dominate marine cloud condensation nuclei (CCN) budget (Fossum et al., 2018), globally more than half of the CCN
form in the atmosphere as a result of new particle formation (NPF) (Gordon et al., 2017). A combination of observations and
modelling work has suggested that marine NPF and growth have a cooling effect on the climate in the North Atlantic region,
especially through aerosol indirect effects (Croft et al., 2021).

The climatic importance of NPF in the marine boundary layer (MBL) has been studied since it was first proposed by Charlson
et al. (1987). The so-called CLAW hypothesis suggested that oceanic phytoplankton could have an influence on climate because
the dimethyl sulfide (DMS) that it emits can be oxidised and form new sulfate aerosol particles that could make the marine
clouds brighter. If DMS emissions by plankton increased with increasing temperature, this could lead to a negative feedback
loop that would slow down climate warming. In practise, the process is less straightforward and the hypothesis has been under
debate (Quinn and Bates, 2011).

More recent research has shown that DMS is not the only possible aerosol precursor gas in marine air. Studies have shown
that amines (Brean et al., 2021) and ammonia (Jokinen et al., 2018) can be important stabilisers for sulfuric acid in particle
formation in marine conditions. In addition to sulfate species, organic compounds can play a big role in the marine CCN budget
(Mayer et al., 2020; Zheng et al., 2020) and iodine, which has been shown to be important for NPF in coastal conditions (He
et al., 2021) is ubiquitous to marine aerosols (Gómez Martín et al., 2021).

Another reason why the CLAW hypothesis has been argued against, is that nucleation would be more likely to occur in the
free troposphere (FT) rather than within the MBL, because sea spray aerosols can act as a sink for aerosol precursors in the
MBL inhibiting NPF. Airborne measurements near Tasmania, and around the Atlantic and the Mediterranean have observed
nucleation occurring in the FT (Clarke et al., 1998; Sanchez et al., 2018; Rose et al., 2015b). Even with nucleation occurring
only in the FT, sulfate originating from DMS could contribute to the CCN budget both by particles transported down from
the FT and by growth of pre-existing particles by sulfate condensation (Sanchez et al., 2018). Recently, Zheng et al. (2021)
showed that around the Atlantic Ocean, NPF can occur in the upper decoupled layer of MBL rather than FT after the passage of
cold fronts. They explain this by convective clouds associated with drizzle and precipitation, both removing large particles and
transporting aerosol precursor gases to the upper decoupled layer. There, between clouds, low pre-existing aerosol surface, high
radiation levels and low temperature favour NPF. Nucleation occurring after cold fronts has been also observed at Cape Grim,
in Tasmania (Gras et al., 2009). These events contributed little to the CCN population, but they increased the concentration of
Aitken mode particles. Other work at Cape Grim has also observed increased particle concentrations at altitudes above 2000 m
compared to ground level and their likely transport to ground level (Bigg et al., 1984).

A coastal field campaign in eastern Australia observed new particle formation and growth of nucleation mode particles in
clean marine air and attributed this to sulfate and organics, hypothesising that the source of these species was likely marine
or coastal (Modini et al., 2009). A recent study showed that at Cape Grim, sulfate from marine biological sources dominates
the CCN population during the summer, but during winter the role of wind generated sea spray aerosols is highlighted (Gras
and Keywood, 2017). Another recent study at Cape Grim showed that part of the secondary organic aerosol in the marine air





masses was derived from isoprene and monoterpene and associated with marine biological activity, but this accounted for less

than 1 % of the total organic aerosol mass (Cui et al., 2019).

Previous studies in the Southern Ocean have shown that at high wind speeds of above 16 ms$^{-1}$, the CCN budget can be dominated by sea spray aerosol, but at lower wind speeds, secondary aerosol can contribute between 92–49 % of the CCN (Fossum et al., 2018). It has also been shown that biological activity is important for marine cloud droplet number concentrations and this has been attributed to both organics and sulfate (McCoy et al., 2015; Mayer et al., 2020). On the other

hand, a recent voyage going around whole Antarctica, did not observe evidence of NPF acting as an important source for CCN (Schmale et al., 2019)

Despite the importance of marine aerosols, most continuous aerosol size distribution measurements are from continental sites and from the Northern Hemisphere (Kerminen et al., 2018; Nieminen et al., 2018). New Zealand is a contrasting environment as it is in the Southern Hemisphere, in the middle of the ocean, far from major pollution sources. It is thus an optimal place

for studying marine air. Despite the optimal location, most aerosol measurements in New Zealand have focused on particulate matter mass concentrations for regulatory purposes and only a few direct observations of NPF in this country exist (Coulson et al., 2016). Baring Head station, located in the southern coast of New Zealand's North Island, has been used for greenhouse gas measurements since 1972. The location of the site was chosen, because it regularly receives clean air masses from the Southern Ocean that have not been in touch with land in days (Stephens et al., 2013).

Previous aerosol measurements at Baring Head have focused on aerosol chemical composition measured with filter samples (Allen et al., 1997; Sievering et al., 2004; Li et al., 2018, 2021), and sulfate aerosol precursors, SO2 and DMS (de Bruyn et al., 2002). Some of the previous work has shown both DMS (Harvey et al., 1993) and non-sea-salt sulfate concentrations in fine aerosols (Li et al., 2018) are higher during the summer than during the rest of the year. Another factor making Baring Head special is the closeness of the biologically active Chatham Rise region (Murphy et al., 2001; Nodder, 1997), described in detail

by Law et al. (2017). One previous study showed that coarse mode aerosols originating from this biologically active area had high alkalinity caused by high calcium concentrations originating from plankton debris (Sievering et al., 2004). This alkalinity enhanced aqueous phase sulfate formation by ozone oxidation in the coarse sea spray aerosol. The plankton was thus not only a source of DMS but also a source of calcium which enhanced SO$_2$ uptake to coarse mode aerosols. As mentioned earlier, on a global level high biological activity has been also connected to larger emissions of particle precursor gases.

We study NPF at Baring Head station in New Zealand over a total period of 10 months covering late autumn, winter, spring and summer. We report typical aerosol concentrations in climate relevant size ranges and statistics for NPF event occurrence and properties and study the conditions favouring NPF. Our focus is on separating the marine signal from land-influenced air masses to study NPF in marine air. Since measurements in the Southern Hemisphere and especially in marine air and in New Zealand are relatively rare, our measurements are highly valuable for the aerosol community.



## 2 Methods

### 2.1 Measurements

Baring Head (41.4083 °S, 174.8710°E) is the longest running measurement station in the Southern Hemisphere for background $CO_2$ measurements. The site was chosen because it enables capturing marine masses coming from the Southern Ocean that have not been influenced by land in several days (Stephens et al., 2013). This is the same reason we chose the station to study aerosol formation in pristine marine air. The site is described by Stephens et al. (2013). Our aerosol measurements were conducted in a separate hut 20 m east and uphill from the main buildings. Our main inlets were 7 meters from the cliff edge at 110 cm height off the ground.

New Zealand sits in a maritime mid-latitude westerly air-flow. The Southern Alps present a barrier to these westerlies but this is broken by the Cook Strait between the North and South Islands. For a range of prevailing synoptic situations which bring westerlies from north-west, through to south-west, air is funnelled through the Strait as a north-westerly or northerly wind at Baring Head located on the north-eastern side of the Strait. These directions result in air masses that have been impacted by the land to the north of the station. There are two main types of situation that cause wind direction to switch and arrive from the south at Baring Head. Firstly, cyclonic situations where low centres pass to the north or across the North Island, drive southerlies or south-easterlies into the Strait. Secondly, there are anticyclonic flows when an anticyclone passes to the south of the South Island or builds in the Tasman Sea to the west of New Zealand. As pressure builds and the ridge moves east, air is deflected around the South Island and arrives at Baring Head as an anticyclonic southerly. These air masses have typically spent several days over the sea and are considered clean marine air. According to Stephens et al. (2013) south and southeast air mass trajectories are observed on average 27 % of the time, being more frequent during the winter than summer. Out of this 27 %, part is still contaminated by land-influences and after filtering, less than 10 % of the data is used for baseline CO2 calculations (Brailsford et al., 2012; Stephens et al., 2013). This section describes the measurements and data analysis methods used specifically for this study.

### 2.1.1 Aerosol measurements

To characterise the aerosol properties at Baring Head, we measured aerosol and air ion number concentrations in different size ranges using several different instruments. A Scanning Mobility Particle Sizer (SMPS) was used to measure the aerosol number size distribution in 10–450 nm during 20.4.2018–13.6.2018 and in 10–500 nm during 12.6.2020–1.3.2021 with a time resolution of 13 min. The fact that the upper limit of the SMPS was lower during the earlier measurement period should not make significant difference for total particle concentrations as particle number concentrations are typically dominated by smaller particles. In fact, during the later measurement period, particles larger than 450 nm contributed only 2% of the number concentration of particles above 100 nm and 0.3 % of total particles. The total length of the SMPS's 1/4" inlet was 312 cm, containing a 73 cm silica gel dryer. To accompany the SMPS, we used one Condensational Particle Counter (CPC) to measure the total concentration of aerosols above 10 nm. From 22.7.2020 to 24.12.2020 the used model was TSI 3010 and





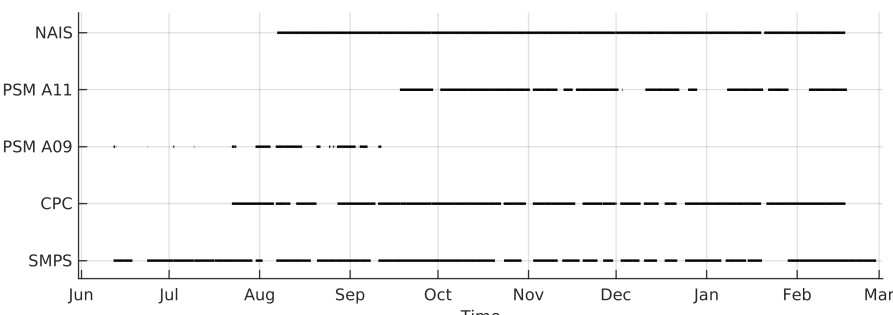

**Figure 1.** Black lines indicate when data is available for each instrument for the 2020-2021 period.

from 24.12.2020 to 17.2.2021 it was model TSI 3760A. The inlet to the CPC was in total 268 cm long and 1/4" thick with the first 103 cm stainless steel and the rest conducting silicon tubing.

To obtain aerosol concentrations in smaller size ranges, we used Particle Size Magnifier (PSM, Vanhanen et al. (2011)). During 12.6.2020-17.9.2020, we used a model A09 with supersaturation fixed at 1 lpm. Then, during 17.9.2020–17.2.2021, we used an A11 PSM in stepping mode with saturation flow rate switching between 0.1 and 1 lpm every 60 s. To assure that changing the instrument did not affect our results, we compared the two PSM's in laboratory over three days. During the intercomparison, the median relationship of model A09 and A11 number concentrations at 1 lpm supersaturation was $\frac{N_{1,A09}}{N1,A11} = 0.93$ (25th and 75th percentiles 0.75 and 1.24) meaning that the concentrations were on average close to each other,

but the relationship varied to both directions. The inlet to the PSM was a 122 cm long 1/4" stainless steel tube. The first 103 cm of this inlet was shared with the CPC to increase the flow rate through the inlet, because increasing the total flow rate from 2.5 to 3.5 lpm decreases the particle losses due to diffusion in an inlet with this diameter.

In addition to SMPS and PSM, we used Neutral cluster and Air Ion Spectrometer (NAIS, (Mirme and Mirme, 2013)) from 7.8.2020 to 28.2.2021. The NAIS measures the size distribution of particles in 2–42 nm and ions in the electric mobility range

from 3.2 to 0.0013 cm$^2$V$^{-1}$s$^{-1}$. The inlet of the NAIS was three meters closer to the cliff edge than the other instruments and at 70 cm height. The data availability from all particle instruments during the 2020–2021 period can be seen in Figure 1 and Appendix Table A1. For 2018, we only had SMPS data and it was available for the whole measurement period of 20.4.2018– 13.6.2018.

### 2.1.2 Ancillary data

In addition to aerosol measurements, we used the station's permanent ozone, radon, and meteorological measurements for temperature, global radiation, relative humidity and wind. The meteorological data can be downloaded from https://cliflo.niwa. co.nz/ (last accessed May 2021). Ozone data are from long-term measurements of the station conducted with Thermo Scientific Model 49i Ozone Analyzer. Radon data were used to assess land-influence. The measurements were done with ANSTO





(Australia, www.ansto.gov.au) designed high-sensitivity site background radon detector (Chambers et al., 12-13 November
145 2014).

As the station is coastal, tides and wave heights can also have an effect on the atmospheric composition. This is why we
also used wave height data from Greater Wellington Regional Council (http://graphs.gw.govt.nz/, last accessed May 2021,
data available only for 2020-2021) and estimated tide heights for Wellington from Land Information New Zealand https:
//www.linz.govt.nz/sea/tides/tide-predictions (last accessed May 2021). The tide height data were interpolated with Piecewise
Cubic Hermite Interpolating Polynomial to obtain data with higher time resolution.

## 2.2 Air mass back trajectories

Air mass back trajectories were calculated with HYSPLIT (Stein et al., 2015; Rolph et al., 2017) for 72 hours with one hour
time resolution. The input meteorological data are from the Global Data Assimilation System (GDAS) model with a one-degree
resolution.

## 155 2.3 Data analysis

### 2.3.1 Air masses

In order to study marine air masses, we separated marine air masses from land-influenced air using air mass back trajectories,
radon concentrations and wind direction. To differentiate between land-influenced and marine air mass back trajectories, we
used landmask code (https://se.mathworks.com/matlabcentral/fileexchange/48661-landmask), last accessed 13 May 2021) to
define how long the air mass back trajectories had spent over land. Only times for which the back trajectories had spent 100 %
of time over the sea were classified as marine. Due to the one hour time resolution of the back trajectories and spatial resolution
of the land data, this method occasionally classifies back trajectories coming from the north as 100 % marine even though they
have to pass over land. This is problematic especially since the area north of the station contains urban areas of Wellington
which act as pollution sources as discussed by for example by de Bruyn et al. (2002). This is why we also used radon and wind
data to separate marine and terrestrially influenced air masses.

Radon (Radon-222) has been previously used at Cape Grim in Australia to identify time periods when air has not been in
contact with land for several days (e.g., Molloy and Galbally, 2014). This is based on radon being emitted from land around
100 times faster than from the sea and having a half-life of 3.8 days. The radon limit traditionally used at Cape Grim is
$100 \ \mathrm{mBqm}^{-3}$ and since the environment is similar to Baring Head, we used the same value to separate between marine and
land-influenced air.

Finally, since we observed some points with radon below 100 mBq m-3 coming from the direction of Wellington city, we
also used wind direction to eliminate these data. Wind direction values accepted for marine air are 120–220°, since this range
has been previously used for Baring Head by de Bruyn et al. (2002). Combining all these criteria we can compare marine air
masses that have not been in touch with land during several days with air masses that have been influenced by land.





In addition to separating between marine and land-influenced air masses, we use the altitude of the back trajectories to estimate whether the marine air masses have been in the marine boundary layer or in the free troposphere. Previous work at Cape Grim has shown that the marine boundary layer is typically mixed up to altitudes of 500–1000 m (Bigg et al., 1984). This is why we decided to use an altitude limit of 500 m to separate between air masses that have likely been within the MBL and air masses that could have come from the free troposphere.

Air masses were also used to identify regions that favoured new particle formation, similarly to the work by Rose et al. (2015c). This was done by combining the air mass back trajectories with the number concentration of negative ions in 2-4 nm. For each time step, we attributed the concentration of ions measured at Baring Head to the full back trajectory path. Then, for a given grid cell, we averaged the resulting concentration by the number of back trajectories that pass through the 1x1°grid cell, which provided a map of the ion concentration occurring when air masses are coming from different grid cells. Only cells that

had at least 10 back trajectories passing through them were accepted.

### 2.3.2 Aerosol data

We combined information from PSM, CPC, and SMPS in order to obtain particle number concentrations in different size ranges. The size ranges we use are 1-10 nm (N1-10) which uses both PSM and SMPS data, between 10 and 100 nm (N10) and above 100 nm (N100) which were both calculated from the SMPS data. We decided to use the SMPS concentrations for 10 nm

particles rather than the CPC, because CPC data were missing for several months.

CPC data were used to check the quality of SMPS data. We compared total SMPS concentrations to total CPC concentrations. All in all, the instruments agreed well, but the SMPS seems to slightly underestimate concentrations with the median of $N_{tot, CPC}/N_{tot,SMPS}$ being 1.51. The differences could be due to higher losses in the SMPS inlet and dryer or lower detection efficiency of SMPS's CPC. This is why we decided to correct the SMPS data by multiplying the concentrations with this value.

Part of the difference could also be due to the fact that the CPC measured all particles above 10 nm while SMPS measured only particles in the 10-500 nm size range, but typically particle number concentrations are dominated by smaller particles, so we consider this negligible. It should be also noted that particle diffusion losses in the inlet and dryer are larger for the smaller particles. This can lead to a bias in the size distribution, and in the case of low nucleation mode particle concentrations, a total loss of particles in the smallest sizes. In addition to the correction made to SMPS data, we removed negative values from

N1-10, as has been done previously for PSM data (e.g., Sulo et al. (2021)). Negative particle concentrations are non-physical and can be both due to differences in instrument efficiencies and measurement times.

SMPS data were also used to calculate the condensation sink formed by the particle population as in Kulmala et al. (2001). This basically represents the surface area of aerosols and describes their ability to act as a sink for condensable vapours.

### 2.3.3 Event classification

As SMPS data are available for the longest total period, we used these data to classify all days into different NPF event classes based on the guidelines by Dal Maso et al. (2005). In this method, particle number size distributions for each day are inspected visually and days are divided to Class I, Class II, undefined and non-event days. Class I and Class II days are days when new



particle formation is clear with the difference that during Class I the new particle mode is clear and determination of growth rates is possible, whereas during Class II events the mode diameter or concentration varies. On non-event days there is no new

particle formation and on undefined days, the new nucleation mode does not grow or there is growth in Aitken mode. Note that the growth of particles in Aitken mode could still be NPF in marine air masses, despite being in the undefined event class with this method that was made using data from a continental site. This will be further discussed in the Results section.

To obtain further information about the events, we also performed event classification based on the method developed by Dada et al. (2018) using NAIS data when it was available. This method is more quantitative than the method of Dal Maso

et al. (2005) and gives us information about the size range of the events as well as event start and end times. The method is an automated procedure based on comparing the daytime ion and particle concentrations to background concentrations during the night. It uses the concentration of ions in 2-4 nm to detect the first step of particle formation and particle concentration in 7-25 nm to determine if the particles grow further or if there is a transported event that started somewhere else and is only observed at the station once the particles have grown. Here, we only used the particle concentration between 7-15 nm, because

our instrument did not work properly for sizes above 15 nm. Even below 15 nm, the concentrations seemed to be somewhat underestimated. Because of this and because the typical particle concentrations at Baring Head are lower than at Hyytiälä, we modified the Dada et al. (2018) algorithm parameters to take this into account. The algorithm has relative and absolute threshold values for both the ions and particles. The relative thresholds refer to comparing the daytime concentrations to background nighttime concentrations and absolute thresholds are fixed values that the daytime concentrations have to exceed in order for

the day to be considered as an event. The relative ion and particle limits that are determined based on the background remained the same, but the absolute thresholds for ion and particle concentrations were lowered. For ions we used $3 \; \mathrm{cm}^{-3}$ instead of $20 \; \mathrm{cm}^{-3}$ and for particles we used $100 \; \mathrm{cm}^{-3}$ instead of $3000 \; \mathrm{cm}^{-3}$. Despite the instrument issues and the different environment, we consider the method reliable, because a comparison to the manual method (see the Results section) seemed good.

### 2.3.4  Growth rates

The rate at which the diameter of a particle mode grows can tell us about the condensational growth of the particles. By comparing the diameter growth rates to other variables, we can find out which factors help the particles grow from nucleation mode to climatically relevant sizes.

Growth rates were determined for all size classes with the method developed by Paasonen et al. (2018). To use the same criteria as Paasonen et al. (2018), we first interpolated the SMPS and NAIS data to 10 minute time resolution. The method

first looks for peaks in the concentration data for each time point and then groups these points based on the diameter at which the peak is observed so that different particle modes are not mixed. Then it goes through each mode and looks for periods where the diameter of the peak is growing. If the growth is monotonic enough and lasts for at least 2 hours, a growth rate is determined as the slope of a linear fitting to the peak points. This slope and the diameter of the growing particles are then saved along with the start and end times of the observed growth period. To further analyse the growth rates, we turn this information

into an hourly time series.



In addition to the automatic method, we defined growth rates manually for the Dal Maso et al. (2005) Class I events. The manual method uses a mode fitting method by Hussein et al. (2008) to find aerosol modes. The user then chooses the geometric mean diameter points in nucleation mode that are related to the event and a linear function is fitted to these points to determine the diameter growth rate during the event. These growth rates were calculated for the determination of formation rates to stay consistent with previous work, such as Nieminen et al. (2018).

### 2.3.5 Formation rates

Particle formation rate is the rate at which particles at a chosen size are formed and it tells us about the intensity of particle formation. To keep our results comparable to other sites, we calculated the formation rates for 10 nm particles, following the same method as Nieminen et al. (2018). The formation rate is defined as

$$J_{10} = \frac{dN_{10-25}}{dt} + CoagS \times N_{10-25} + \frac{GR}{\Delta d_p} N_{10-25}, \tag{1}$$

where $\frac{dN_{10-25}}{dt}$ is the change of concentration in 10-25 nm particles, CoagS is the coagulation sink calculated for 15 nm particles, and the last term defines the growth losses out of the size range.

## 3 Results

To give an overview of new particle formation at Baring Head, we begin this section by classifying all days to event and non-event days with traditional methods and characterise these events. Then we look at aerosol concentrations in more detail and study aerosol formation and growth. Finally, we focus on the special characteristics of new particle formation in marine air masses.

### 3.1 New particle formation events

#### 3.1.1 Event occurrence and characteristics

To get a general overview of how common new particle formation is at Baring Head, all days with SMPS data were classified with the criteria by Dal Maso et al. (2005). Over all, 10.9% of the days were Class I events. Additionally, 12.1% of the data were classified as Class II events making the total average event frequency 23.0%. 32.3% of the days were classified as undefined, leaving 44.8% as non-events. Even though most previous studies of NPF frequencies have been made for continental Northern Hemisphere sites, our numbers are comparable to other remote sites (Nieminen et al., 2018). NPF events in New Zealand have been previously observed in Auckland at a site that was 20 km from the sea (Coulson et al., 2016). They hypothesised that particle formation was favoured by low pre-exiting aerosol concentrations and particle forming vapours could have been a combination of biogenic emissions from both the ocean and a forested area and urban precursors, but no data confirming this hypothesis were available. Similar factors likely played a role at Baring Head.





The seasonal cycle of the fraction of event classes (Fig. 2), shows that the highest event frequency is observed during late
spring in November (38.1 %). The lowest event frequency (14.3 %) is observed in December, but this month contains many
undefined days, making the fraction of non-events no higher than most months. Both the lowest numbers of Class I events
and highest numbers of non-event days were observed in May-June, indicating that particle formation was less frequent during
the winter. The data from April were not included in the seasonal cycle as we only had 10 days of data from April. The only
two coastal sites at similar distances from the equator in the study of Nieminen et al. (2018) were Mace Head in Ireland and
Finokalia in Crete. The lowest event frequencies at both of those stations were observed during the winter, with 6.5% for Mace
Head and 16.3% for Finokalia. Highest values were observed during the spring with 29.3% for Mace Head and 36.6% for
Finokalia. Our results are similar to these stations, especially to Finokalia which has a more similar distance to the equator.

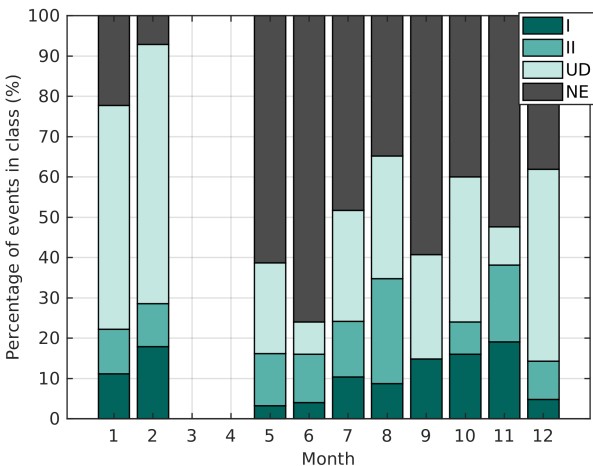

**Figure 2.** Bar plots show the fraction of each event class during each month. I stands for Class I, II for Class II, UD for undefined and NE
for non-event days.

Previous studies at Baring Head and its surroundings have observed more DMS (Harvey et al., 1993) and more non sea salt
sulfate (Law et al., 2017; Li et al., 2018) during the late spring and summer. This could be one of the factors increasing NPF
frequency during this time. As photochemistry is important for particle formation, another possible factor is more favourable
meteorological conditions during the summer. In addition to longer days, the summer at Baring Head is characterised by less
southerly winds than during the winter season (Stephens et al., 2013) and southerly winds are often related to cloudy, windy,
and rainy weather which would inhibit NPF.

Figure 3 illustrates the time that the air mass had spent in marine air and marine free troposphere during each of the event
classes. It should be noted that this analysis contains all the data between 8–15 h and while that corresponds to typical event
times, the event times vary day to day (see Fig. A2). This analysis shows that most of the Class I events are likely influenced by
some time spent over land, with the median time that the air masses had spent over land during those days being 13 h. Many of
the Class II and undefined days had also land influence, with the median time spent over land being 7 h. Non-event days, on the





other hand, were more common when the air mass had spent all of its time in marine air. This shows that NPF events classified
with the Dal Maso et al. procedure are relatively rarely found in pure marine air. Observing particle formation in air masses
that come from the sea and then cross over land is not surprising since marine air is typically characterised by low particle
concentrations and the sources of particle precursor vapours over land are typically higher than over the oceans. If we look at
the time spent in the marine free troposphere (Fig. 3b), we see that during events, the air masses were more likely to have spent
time in the free troposphere than they were during non-event days. This could be explained both by lower pre-existing particle
concentrations in the air that came from the troposphere or the transport of particle precursor vapours or particles themselves
from higher altitudes. Here, it should be noted that the division to free tropospheric air was only done based on a fixed threshold
for the altitude of the air mass back trajectories even though the start height of the free troposphere can be variable.

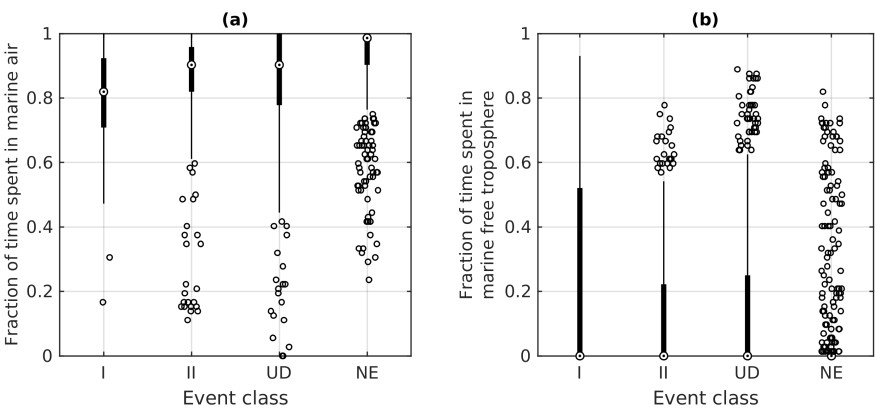

**Figure 3.** Fraction of time the air mass has spent in (a) marine air and (b) marine free troposphere between 8-15 h during days classified to
Class I (I), Class II (II), undefined (UD) and non-event (NE) days. The circles are the median concentrations for each month, black boxes
mark 25th and 75th percentiles and rest of the points are outside this range.

In order to obtain more information about the NPF events, we used a second classification method developed by Dada et al.
(2018) which characterises the events with more detail. This classification is based on NAIS data, meaning that data were not
available for most of the winter months (see methods). With this method, we observed regional events that start from ions of
2-4 nm and continue to grow past 7 nm on 26.2% of the days. 15.4% of the days were classified as transported events where
the first steps in ions are not seen, but a nucleation mode is observed. Only 6.7% of the days were ion burst days during which
ions in 2-4 nm appear but do not grow to larger sizes. The rest (51.8%) were non-events.
Comparing these results to the classification by Dal Maso et al. (2005) (Appendix Fig. A1) showed that all the Class I events
and most of the Class II events were regional or transported events, which is in line with the fact that this class requires clear
growth in nucleation mode. Two of the Dal Maso et al. non-events were classified as ion bursts, which is reasonable as ion
bursts appear in a size range smaller than that used for the Dal Maso et al. classification. This shows that Dal Maso et al.
classification might miss the initial steps of NPF if no further growth occurs. Many of the undefined days are classified as





non-events by the Dada et al. (2018) method, but this is explained by the fact that the undefined class includes days where
growth in pre-existing Aitken mode was detected.

We used the method by Dada et al. (2018) also to define event start and end times (Appendix Fig. A2). This definition is
based on the time evolution of the concentration of ions in the 2-4 nm size range and thus tells about the first steps of NPF. The
average event duration with this method was only 3 hours. Typical start times were around 8-10 h in the morning and typical end
times around 13-15 h in the afternoon. It should be noted that particles might continue to grow in larger size ranges even after
small ions are no longer detected, meaning that this method might underestimate the total event duration. One weakness of this
method is also the fact that, as it is using the night time concentrations as background concentrations, nighttime events could
not be detected with this method. Based on visual inspection of the data, no clear NPF events occurred during the nighttime,
but few potential night time ion burst events did happen and as shown later in Section 3.5.3, we saw nighttime increases in
sub-3 and sub-10 nm particle concentrations.

To see more quantitatively how many of the events occur in marine air, we checked the percentage of time that the back
trajectories were marine during the events (Appendix Fig. A3). This was calculated for regional and ion burst events, as start
and end times are only defined for these event classes. Air masses during all events had spent over 50% of time over sea. For 12
events, the air masses had only spent time in marine air according to the back trajectory calculation. This is 18.75% of the total
RE and IB events and 6.15% of all days for which NAIS data were available. Half of these fully marine events were classified
as regional events and half as ion bursts. However, out of these 12 events, only one met the other criteria for clean marine air
and had radon below $100 \ \mathrm{mBqm^{-3}}$ and this event was surrounded by land-influenced periods. This means that during most of
the events, the air had recently passed the southern tip of North Island and these events could have some land-influence. Based
on this, events classified with the Dada et al. criteria in completely clean marine air seem rare, but on the other hand, only 7.3
% of our measurement time was classified as clean marine air and there were only 26 days that had more than 30 % of data
in clean marine air, so longer time series would be required to get more statistics on the importance of NPF in clean marine
air. Also, a classification specific for NPF in clean marine air masses that take into account the low concentrations, potential
nighttime cluster formation and slow growth might be necessary in the future, as will be discussed in Section 3.5.

### 3.1.2 Factors favouring NPF event occurrence

Since this is the first time NPF events have been observed at Baring Head, we compared the meteorological conditions occurring
during event and non-events to understand which conditions favour event occurrence. In addition to basic meteorological
variables (global radiation, temperature, relative humidity (RH) and wind speed), we also compare typical condensation sinks
and ozone levels.

Figure 4a clearly shows that global radiation levels are high during events and lower during non-events. This is no surprise,
since photochemistry is likely to play an important role in particle formation. For temperature (Fig. 4b), we can see that during
event days, the temperature is low in the morning but then increases clearly over the day. This shows that the start of new
particle could be favoured by cold conditions. The daytime increase can be explained by sunny conditions warming the air
during the day. This trend is similar to what Jokinen et al. (2021) observed in Northern Finland.





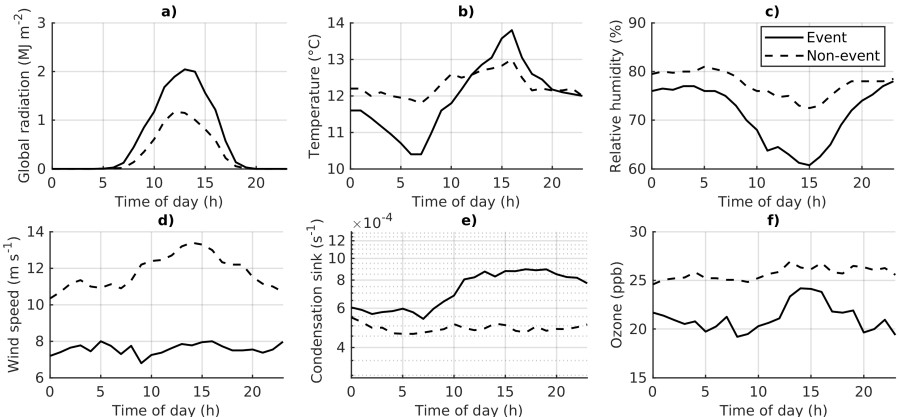

**Figure 4.** The diurnal cycles of different meteorological variables, condensation sink and ozone during event and non-event days.

With relative humidity, lowest values are observed during events and highest during non-events (Fig. 4c). This could be expected based on the results with radiation, since high relative humidity typically correlates with low level clouds which

block radiation (Hamed et al., 2011). Previous work at a remote coastal site in Australia has also shown that new particle formation occurred when radiation levels were high and RH was low (Modini et al., 2009).

Figure 4d shows that events are favoured by low wind speeds. Wind speed can be related to both the amount of sea spray aerosol in the air and the time that the air has spent over land. At high wind speeds, waves are typically higher and produce more sea spray aerosols which can act as a sink for aerosol forming condensable species. On the other hand, at Baring Head,

the wind speeds are typically lower when the air mass has spent more time over land and we saw earlier, events are more likely to occur when the air mass has spent some time over land.

With condensation sink (CS), we can see that on non-event days the CS is low while on event days CS is slightly higher and increases clearly during the day. This shows that new particle formation can likely increase the CS. Having higher CS on event days is opposite to what has been seen for example in Southern Finland (Dada et al., 2018) and the common assumption that

higher condensation sink would prohibit NPF by acting as a sink to particle precursor vapours. Our results are nevertheless reasonable since at Baring Head the events occurred primarily over land whereas non-event days had primarily marine air. Over land, the sources of aerosol precursor species seem to be more intense than over land. This can set off particle formation if the CS has not yet increased too much and meteorological conditions are favourable. Similar results have been seen at mountain sites (Boulon et al., 2010; Rose et al., 2015a).

Finally, for ozone (Fig. 4f), the levels are lower during event days compared to non-event days. The data should be studied further to understand whether this is a question of ozone chemistry influencing NPF or just a difference between land-influenced and marine air masses.





## 3.2 Aerosol concentrations

Since these are to our knowledge the first measurements of aerosol particle number concentrations starting from 1 nm in New
Zealand and the longest data set of aerosol number concentrations at Baring Head, we explore the seasonal and diurnal cycles
of particle number concentrations. The cycles can also give us information about the factors controlling aerosol concentrations
at Baring Head. The size ranges we use here are 1-10 nm (N1-10), 10–100 nm (N10-100), and above 100 nm (N100). N1-10
typically consists of new particles formed in the atmosphere, N10-100 can contain both secondary particles that have grown
from sizes below 10 nm and primary particles and for N100 primary particles are likely more important than for the smaller
particles.

All the data were divided into marine and land-influenced data points based on air mass back trajectories, radon, and wind
direction. Aerosol number concentrations in all used size ranges were lower in marine air masses than in land-influenced air
masses, with median (25th-75th percentile) N1-10, N10-100, and N100 nm being 270 (100–730), 580 (360–890) and 110 (80–
180) cm$^{-3}$ in marine air and 710 (300–1630), 1020 (540–2010) and 170 (100–280) cm$^{-3}$ in land-influenced air, respectively.
This is reasonable as marine air masses are typically cleaner than continental air masses.

A previous voyage conducted east of Baring Head observed particle concentrations of $534 \pm 338$ cm$^{-3}$ in clean marine air
and $1122 \pm 1482$ cm$^{-3}$ during land influence (Law et al., 2017). Our results are within the same range. Voyages west of New
Zealand have observed N10 of 681 (388–839) cm$^{-3}$ at latitudes similar to Baring Head (Humphries et al., 2021), which is
also in line with our results. Out of other coastal sites, previous work at Mace Head by Dall'Osto et al. (2011) saw N10 of 327
cm$^{-3}$ in open ocean air and 1469 cm$^{-3}$ during open ocean nucleation. During coastal nucleation and anthropogenic influence,
the numbers were higher (nucleation 2548 cm$^{-3}$, anthropogenic 1580 cm$^{-3}$). Our numbers are between their two open ocean
classes, which is logical since coastal sources do not seem to be important at Baring Head (see Section 3.5.1).

Seasonal cycles can be observed for the particle concentrations in all the size ranges (Fig. 5). For the smallest size range
of 1-10 nm particles, we only have enough data from 7 months (in June and July 2020 data available only 2% and 11% of
the time), but we can still see that the concentrations are the lowest in both land-influenced (Fig. 5a) and marine (Fig. 5d) air
masses during late winter and early spring (August-September) and higher later during the spring and summer. The differences
between months are more significant in the marine air masses than in land-influenced air, but this could be partly explained by
the fact there are less marine data. The median monthly particle concentration in marine air is only 64 cm$^{-3}$ in August and
increases significantly during the spring, reaching the highest median of 637 cm$^{-3}$ in October. During the spring (September-
November), N1-10 comprises 29 % of the total particle number in marine air, indicating that nucleation likely occurs in these
air masses with a large enough frequency and intensity to influence the total aerosol particle concentration. This also implies
that classification of NPF events with the classical criteria (Dal Maso et al., 2005; Dada et al., 2018) originally designed for
a continental site might not be suitable for the detection of nucleation in a remote marine environments. The seasonal cycle
can be related to both biological sources of particle precursors and meteorological conditions favouring nucleation during the
spring and summer.





In the second size bin of 10-100 nm (Figs. 5b and e), the lowest concentrations are observed during June and July in both air mass classes. Similar seasonal cycle for Aitken mode particles in marine air has been observed before at Cape Grim (Bigg et al., 1984). Again, during the winter we are less likely to see new particle formation. Another reason that could decrease particle concentrations in this size range more in winter compared to the summer is losses due to more wet deposition by rain.

For particles larger than 100 nm, the seasonal cycles are less clear, but the smallest medians are again observed during the winter and the highest during summer and late autumn. The fact that the cycle is less clear than in Aitken mode likely indicates that primary emissions, such as sea salt in the marine air and anthropogenic emissions in the land-influenced air, are more important relative to secondary particle formation in this size range compared to smaller sizes. The higher summer values can again be explained both by meteorological conditions and more active biological source during the summer. As mentioned

earlier, previous work at Baring Head has shown that non-sea-salt sulfate concentrations in fine aerosols are higher during the late spring and summer (e.g., Li et al., 2018; Allen et al., 1997) and this secondary sulfate could increase particle concentrations in climate relevant sizes as well.

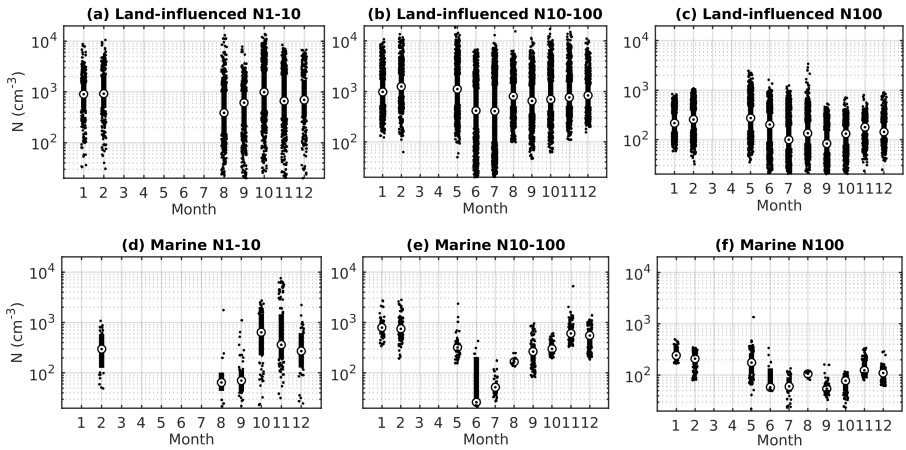

**Figure 5.** Seasonal cycles of particle number concentrations (N) in 1-10 nm, 10-100 nm and above 100 nm during land-influenced (a-c) and marine (d-f) land masses. The circles are the median concentrations for each month, black boxes mark 25th and 75th percentiles and rest of the points are outside this range.

Looking at the diurnal cycles of particle concentrations can give us more information about the processes controlling particle concentrations. The clearest diurnal cycles can be seen for land-influenced N1-10 and N10-100 which both increase during

the day (Fig. 6a-b). Median N10-100 is below $700 \ \mathrm{cm^{-3}}$ in the morning and increases to above $1000 \ \mathrm{cm^{-3}}$ in the afternoon. This is likely explained by particle formation during the day. N100 has a similar but weaker cycle, with median concentrations increasing during the day by less than 35 % compared to early morning hours. The fact that the concentrations in all size ranges increase steadily through the day, rather than for example having peaks during rush hours indicates that particles could grow past 100 nm with photochemistry during the day.





In marine air, the cycles are less clear and the concentrations vary less, especially in sizes past 10 nm. This is partly due to the fact that we have a lot less data from clean marine air masses, but it could also indicate that photochemistry and secondary aerosols play a smaller role in marine air than in land-influenced air.

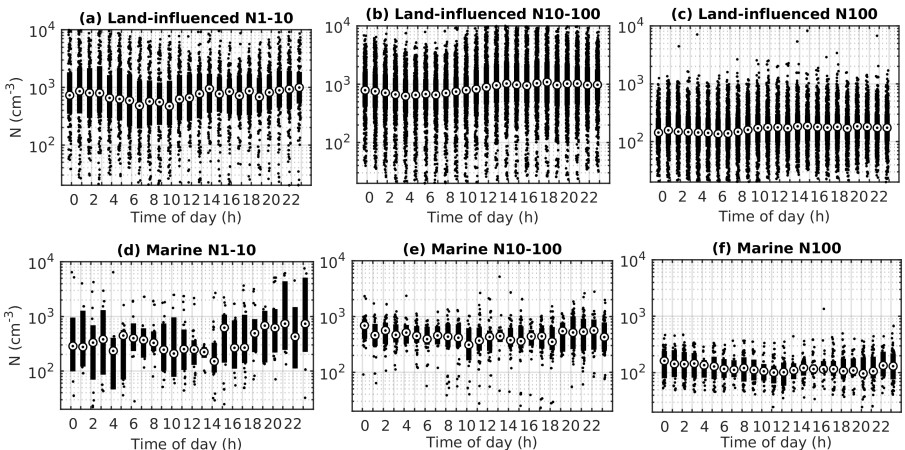

**Figure 6.** Diurnal cycles of particle number concentrations (N) in size ranges of 1-10 nm, 10-100 nm and above 100 nm separated to marine and land-influenced air. The circles are the median concentrations, black boxes mark 25th and 75th percentiles and rest of the points are outside this range.

Finally, we look deeper into the effect of land influence on particle concentrations. Figure 7 illustrates the effect of time spent over land on particle number concentrations. While time spent over land does not explain all of the variations in particle concentrations, the connection is clear. For N100, particle concentrations increase during the first hours spent over land, but eventually they stabilise, with changes after 40 h being small. For sub-100 nm particles, there is an optimum time spent over land that is most favourable to particle formation. We used a second order polynomial fit to describe this time evolution. While the fits are not very good, and we have $R^2$'s of 0.03, 0.04 and 0.006 for N1-10, N10-100 and N100, respectively and root mean square errors of 1960, 1660 and 620 $\mathrm{cm}^{-3}$ for N1-10, N10-100 and N100, respectively, the fits describe the general behaviour of particle concentrations. Both N1-10 and N10-100 reach their maximum concentration after the air mass has spent around 37 h over the land, but N100 keeps increasing and according to the fit it would only start decreasing after 101 h. This is logical, since when the air mass arrives from the sea to land, particle concentrations and the condensation sink are low which favours new particle formation. This increases the sub-100 particle concentration, but after a while, the concentrations get saturated because the condensation sink increases and starts to limit NPF.

In accumulation mode, the decrease is slower and continues a longer time. This is likely due to a larger fraction of the accumulation mode particles being primary particles. Primary particle emissions would not be suppressed by increasing condensation sink the same way secondary particle formation is. While the number increase in accumulation mode is slower than for the smaller modes, the concentration is a lot lower to begin with and it doubles in approximately one day. The concentration of accumulation mode particles is very important for cloud formation, because in this size range, aerosols are likely to activate





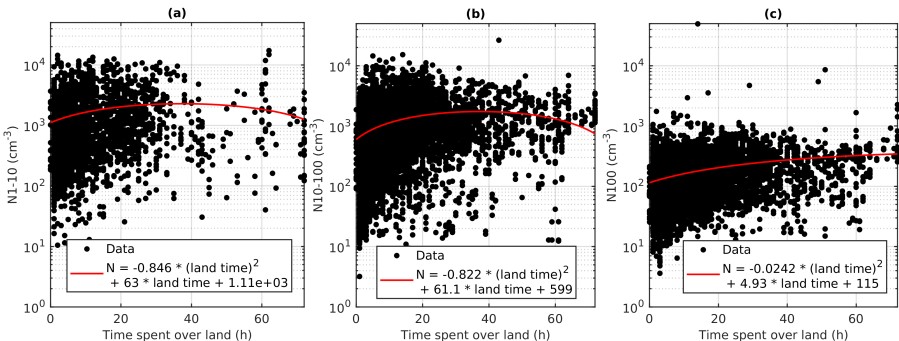

**Figure 7.** Number concentrations of particles in 1-10 nm (a), 10-100 nm (b) and above 100 nm (c) as a function of time spent over land and the fits used to describe the time evolution in red with the equations in the legends.

to CCN. Previous work has shown that doubling cloud droplet number concentration can nearly double the cooling effect of low-level marine clouds (Rosenfeld et al., 2019). Aerosol production over New Zealand is thus likely to play an important role in regional cloud formation over New Zealand and its surroundings.

### 3.3 Growth rates

To understand secondary aerosol formation at Baring Head better, this section shows the behaviour of aerosol diameter growth
rates. Particle growth is important also because larger particles can in general activate as CCN at lower supersaturation levels (e.g., Kerminen et al., 2012). In total, the automated method calculated 652 growth rates out of which 197 started in nucleation mode, 356 in Aitken mode and 99 in accumulation mode. The average growth duration was 3 h 17 min.

The median growth rates were 1.6 (25th-75th percentiles 0.6–2.6), 1.6 (0.7–2.9) and 3.6 (1.6–6.2) nm/h for nucleation, Aitken and accumulation modes, respectively. A global study looking into nucleation mode growth rates saw slightly higher
values at coastal sites with annual median growth rates of approximately 2.5 and 4 nm/h for Mace Head and Finokalia (Nieminen et al., 2018), respectively, which is reasonable since our site is more remote. Growth rates being higher for larger sizes has been previously observed for a boreal forest and around the Atlantic Ocean (Paasonen et al., 2018; Burkart et al., 2017). In those studies the increase of growth rates at larger sizes was explained by the role of semivolatile species, which are involved at a later stage of the growth.

If we divide the growth rate data set to fully marine and land-influenced growth rates, only 70 of the growth rates fit our criteria of clean marine air with fully marine back trajectories, average radon during growth below 100, and wind direction between 120–220°. Out of these 17 were in nucleation mode, 39 in Aitken mode, and 12 in accumulation mode. This means that we observed growth starting from nucleation mode 16.2 % of the time in clean marine air. For Aitken mode, this percentage was 26.4 %, and for accumulation mode 7.1 %. For marine air only, the median growth rates were 0.7 (0.4–2.0), 0.6 (0.1–2.3)
and 2.5 (1.2–3.7) nm/h, for nucleation, Aitken and accumulation modes, respectively. The growth rates are lower in marine air than in land-influenced air which can be explained by lower concentrations of particle growing precursor species. A previous





study looking into nucleation and Aitken mode growth in open ocean air at Mace Head saw typical growth rates of 0.8 nm/h (O'Dowd et al., 2010), which is similar to our results. These results show that even if we did not observe classical NPF events in clean marine air, particle growth starting from the nucleation mode is still frequent, meaning that new particle formation

may have occurred but may have not been classified as a NPF event in the conventional classification designed for continental data. Moreover, we also observe the growth of larger particles frequently, meaning that secondary aerosol formation can be important for the marine CCN budget.

The diurnal cycles of growth rates in different modes (Fig. 8) were made based on a half an hourly time series made out of the growth rate data. It shows slightly higher nucleation mode growth rates during the day compared to before 9 h. This is

logical since photochemistry can produce vapours that participate in particle growth. For Aitken mode particles, growth rates show morning (5–6 h) and early evening (16-18 h) minimums with median growth rates being highest during the day and late evening. In accumulation mode, the median values increase over the morning with maximums around midday and late evening. One possible factor explaining the higher growth rates towards the end of the day could be that the particles have grown to larger sizes by then and as mentioned before, larger particles can grow faster. Although both nucleation and Aitken modes have

relatively high values during the summer (January), no well-defined seasonal cycles were seen for the GRs of any of the modes (Appendix Fig. A4). There are so few growth rate values for marine air that the diurnal and seasonal cycles are not reported separately for marine and land-influenced air.

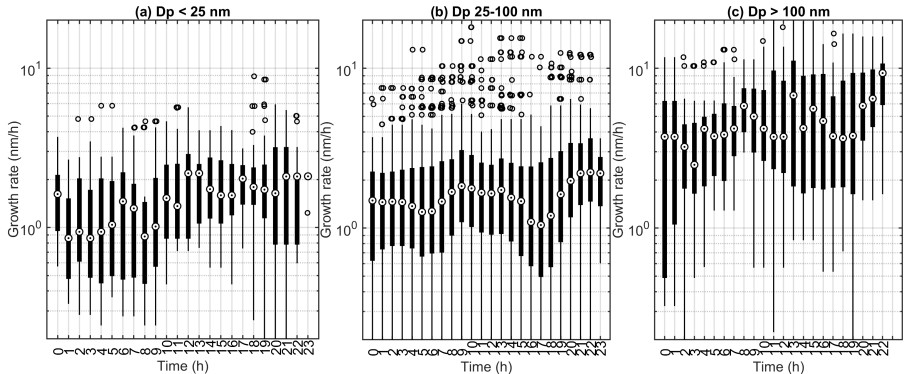

**Figure 8.** Diurnal cycles of growth rates divided by the diameter (Dp) of growing particles. The circles are the median concentrations, black boxes mark 25th and 75th percentiles and rest of the point are outside this range.

### 3.4 Formation rates

Formation rates describe the intensity of particle formation and are proportional to the concentrations of particle forming

chemical species. We found in total 28 Class I event days for which calculation of growth rates with the traditional mode fitting method was possible. The median formation rate of all these events was 0.18 (0.07–0.40) $\mathrm{cm^{-3}s^{-1}}$. Our values fall in the same range as the values reported by Nieminen et al. (2018) for rural sites. In their study, the only coastal sites were Finokalia and





Mace Head and their annual medians were around 0.35–0.4 $\mathrm{cm^{-3}s^{-1}}$, which is approximately the double of ours. Again, this is presumably due to the remoteness of the Baring Head site. None of the events for which formation rates were calculated

met our criteria for clean marine air. For four events, the air had spent over 95 % of the time above oceans and for these events formation rates were below 0.12 $\mathrm{cm^{-3}s^{-1}}$, which is below the total median, again supporting the interpretation that the amount of particle forming and growing precursor vapours is lower in the marine atmosphere compared to land-influenced air. However, no correlation was found between formation rates and time spent above land or radon concentration.

### 3.5   Marine new particle formation

The unique location of the Baring Head station enabled studying clean marine air masses that had spent several days over open ocean before arriving at the station. This is why we were especially keen on studying secondary aerosol formation in these marine air masses. Even though most of the classical new particle formation events were observed in land-influenced air masses, the previous sections showed that sub-10 nm particles and particle growth from nucleation mode were frequent in the marine air masses. This section studies the potential sources of marine aerosols in more detail and shows examples of new

particle formation in clean marine air.

#### 3.5.1   Coastal effects

At some coastal sites, such as Mace Head, coastal sources can play a large role in NPF (e.g., Dall'Osto et al., 2011), because when coastal macroalgae are exposed to air, they can emit particle producing iodine species. To see if that is the case at Baring Head, we studied the relationship of negative 2-4 nm ions and tide height (Fig. 9). We decided to use this ion concentration,

since it marks the first steps of particle formation. We used only data between sunrise and sunset since photochemistry would likely be important. We coloured the data in Fig. 9 with wind direction to see if more particles are produced from some direction, for example, if there is more macroalgae that produces particle precursors on one side of the station. No correlation was observed between ion concentration and tide height (R = 0.0092, p = 0.5082), which indicates that coastal sources are likely not important for particle formation at Baring Head. The wind direction colouring also shows no effect on ion concentrations.

The vertical 'lines' in the tide height data in the plot are due to the 0.1 m resolution of the tide height data. The lack of connection to tides can be partly explained by the fact that at Baring Head, the tide height varied by less than 1.5 m, whereas in Galway Bay, where Mace Head is located, the water level can vary by up to 4 m (Ren et al., 2015). Iodine emissions are also very different for different algal species (Carpenter et al., 2000) and we do not know which species are present close to Baring Head.

#### 3.5.2   Regions favouring particle formation

To obtain more information about geographical locations that could favour particle formation, we used the air mass back trajectories to see if back trajectories coming from some areas would be more likely to form particles than others. This could be the case for example if some areas of the ocean were more biologically active and produced more particle emitting precursor



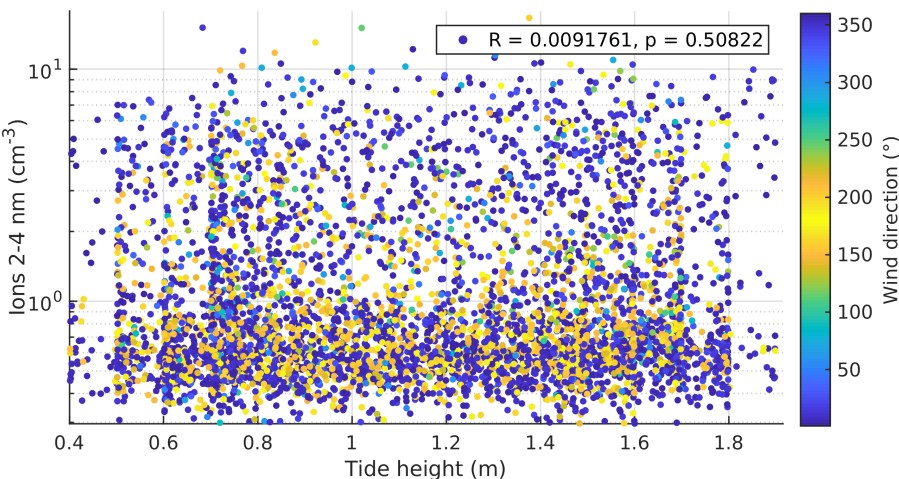

**Figure 9.** The number concentration of ions as a function of tide height coloured with wind direction. Only data after sunrise and before sunset are included.

species. A similar method has been earlier used by Rose et al. (2015c). Figure 10a shows all the back trajectories weighted by 2-
4 nm ion concentrations. This parameter was chosen because it often indicated the start of NPF. The highest ion concentrations
are observed when the air masses come from Tasmania and the sea east of Tasmania. This could be explained both by the
transport of particle precursors from Tasmania and the fact that when the air mass comes from this direction, it typically has to
cross over the Wellington region before arriving at the station.

Apart from the high concentrations coming from Tasmania, some higher concentrations can be seen just north of New
Zealand and in some patches over the Southern Ocean. The area north of New Zealand could be related to air masses pass-
ing through the North Island. Patches with higher concentrations in the south could be related to air transport from more
biologically active areas.

Figure 10b follows the same concept as Figure 10a, but uses only back trajectories that were classified as fully marine and
come from the southerly wind sector (120–220°). We did not use the radon criteria for this figure since there were too little data.
Now the highest ion concentrations appear for the most northwestern back trajectories. This could be explained by these air
masses crossing the coast of the South Island. Apart from that area, the trends are not too clear, meaning that the geographical
area from which the air masses come from over the ocean is likely not important for particle formation when looking at data
integrated over several seasons.

### 3.5.3  Example of marine new particle formation

Since both sub-10 nm particles and particle growth starting from nucleation mode were frequent in clean marine air masses,
but these did not classify as traditional NPF events, marine new particle formation should be studied with different criteria than
classical NPF. Here, we look deeper into some of these new particle formation and growth events in clean marine air and the





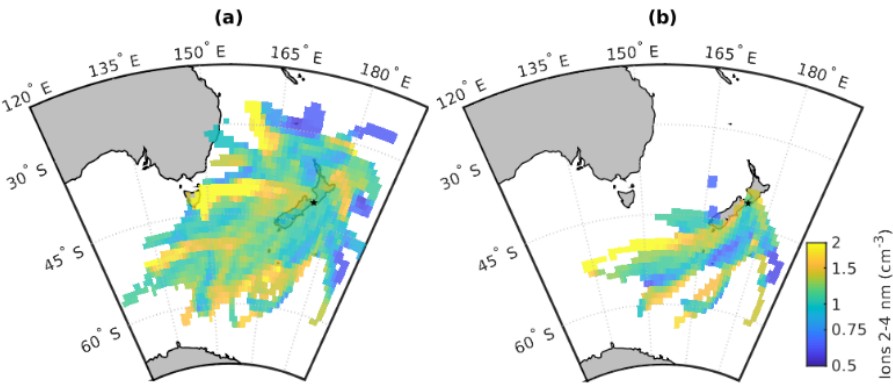

**Figure 10.** Source regions of 2-4 nm ions for all data (a) and only marine data (b).

conditions prevailing during the events. This way, we can better understand the processes driving new particle formation and growth in clean marine air.

Figure 11 shows an example of several growth periods, occurring in clean marine air during the 9th and 10th of November 2020. Part of these growth periods start from the nucleation mode and part from larger sizes. During this time, the aerosol size distribution is largely dominated by two modes, one centred around 20-30 nm and an other around 90-140 nm. The whole period is characterised by high wind speeds (> 18 $\mathrm{ms}^{-1}$) and temperatures below 11 °C (Fig. 12). Significant wave heights vary between 1.7–5 m and clouds are likely present for most of the period. On the 9th, the fact that global radiation levels are

elevated but do not follow a clear parabolic shape likely indicates the presence of scattered clouds. On the 10th, on the other hand, global radiation levels remain below 0.5 $\mathrm{MJm}^{-2}$ during the whole day, indicating that the day was very cloudy. This is supported by the relative humidity being above 80 % from midnight to late afternoon. All air mass back trajectories during this period originated from the ocean south west of New Zealand (Appendix Fig. A5). The heights of all the back trajectories remained below 400 m, indicating that the air masses had likely spent the past three days within the marine boundary layer.

The 9th of November day was not classified as an NPF event by the Dada et al. (2018) method and it was only an undefined day with the Dal Maso et al. (2005) criteria, because nucleation mode particle concentrations were already high at the beginning of the day. However, there are several growth periods, most of which occur between midday and midnight of the 9th. During this time, wind speed and wave height are lower than before and after this period, which likely decreases the sink of condensing vapours. Compared to the second day, this day is also less cloudy, which enhances the photochemical processes. On the second

day (November 10th), we see only a very weak growth in Aitken mode in the afternoon, which is not surprising since the day is very cloudy and there are high waves and wind speeds.

While the lowest size bin of SMPS data remains low for most of the time, in the early hours of November 9th, the concentrations of sub-10 nm and even sub-3 nm particles peak clearly (Fig. 11b). The fact that the sub-10 nm particles reach concentrations as high as 1600 $\mathrm{cm}^{-3}$ during this clean marine air period indicates that nucleation can occur within the marine

boundary layer and at nighttime. When the concentration in the sub-10 nm size classes decreases, we can see some particles





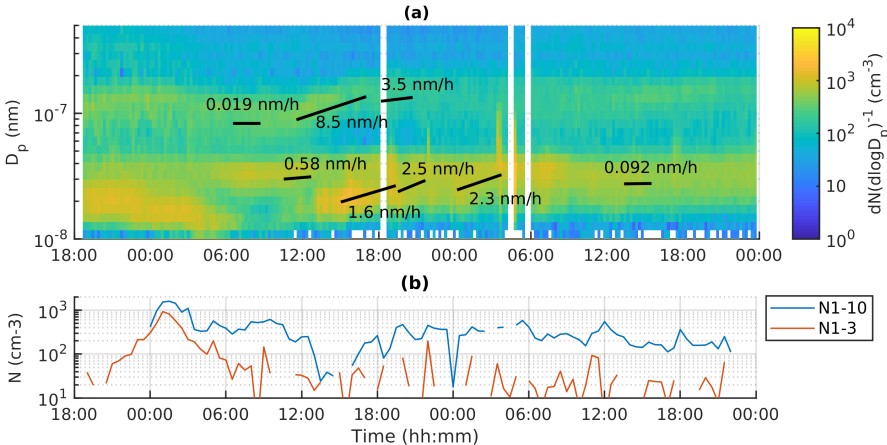

**Figure 11.** An example of particle size distribution and observed growth rates (a) with the 1-3 nm and 1-10 nm particle number concentrations observed with PSM (b) from November 8th at 18:00 to end of November 10th 2020. The vertical white stripes in the size distribution correspond to land-influenced periods that were not included here.

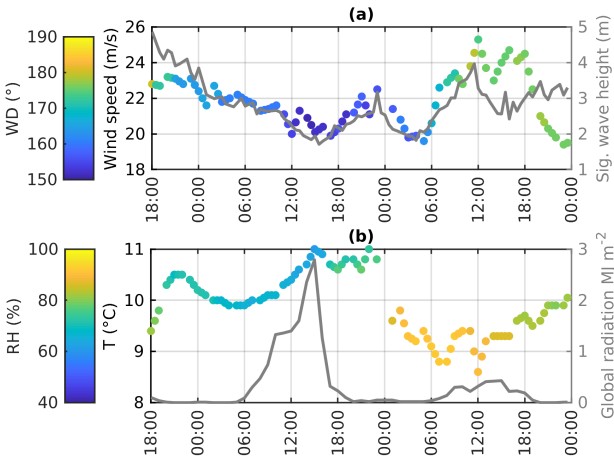

**Figure 12.** Summary of meteorological conditions and wave height from November 8th at 18:00 to end of November 10th 2020.

in the lowest size bin of SMPS data, indicating that these particles grew past 10 nm. After 6 h, however, this mode seems to weaken. This coincides with the sunrise, so one possible explanation is that when the sun rose, boundary layer height increased and the particles concentrations were diluted. The N1-10 remains relatively high until 12–13 h, when we see the appearance of a clearer growing nucleation mode. It is thus possible that the particles that were formed after midnight survived and only started growing in nucleation mode in the afternoon when there was more radiation. It should be noted that the particle concentrations in the lowest size bins of the SMPS data are likely underestimated because of diffusion losses in the inlet, the dryer and the instrument itself. This is to some extent true also for the sub-10 nm and especially sub-3 nm particle concentrations





although the inlets to the PSM and CPC were shorter than the SMPS inlet and did not contain a dryer. This example illustrates that event though this day is not classified as a typical 'banana' type event (e.g., Heintzenberg et al., 2007), new particles likely

formed and grew within the marine boundary layer.

Simultaneous growth of the smaller and larger modes during the 9th shows again that growth is faster for larger particles. The highest growth rate of 8.5 nm/h observed between 12 and 18 h on the 9th could be related to cloud processing and aqueous phase processes increasing the particle size. This is supported by a decrease in the concentration of 60-90 nm particles below the growing mode. Particles in this size range correspond to sizes in which the particles could have been activated into cloud

droplets and grown due to cloud processing leading to a so called Hoppel minimum (see e.g. Noble and Hudson (2019)).

While it is possible that some of the nucleation mode particles that we see growing in the clean marine air come from sea spray (Schwier et al., 2015; Forestieri et al., 2018), observing particles below 10 and even below 3 nm strongly indicates that nucleation can also occur in the marine boundary layer and the freshly nucleated particles can grow to larger sizes. Previous studies at Mace Head, in Ireland, have observed growth events similar to ours in open ocean air masses but whether these

particles originated in the marine boundary layer or free troposphere was not certain (O'Dowd et al., 2010; Dall'Osto et al., 2011). Since our observations also contain measurements of aerosol particles in the size range of freshly nucleated particles, our work shows evidence that nucleation could be occurring within the marine boundary layer. One key message from our work is that marine secondary aerosol formation should not be studied with the same criteria as continental new particle formation.

### 3.5.4    Factors favouring new particle formation in marine air

As shown in this paper, particle formation in marine air masses does not follow the traditional event criteria. This is why, in addition to the analysis in Section 3.1.2, we compared meteorological conditions in marine air with high and low concentrations of sub-10 nm particles to understand the factors driving marine particle formation. Here, we separate the data in marine air masses to times when N1-10 is less than or greater than $500 \ \mathrm{cm}^{-3}$. This somewhat arbitrary limit was chosen because in the example figure (Fig. 11), the clearest peak in N1-10 exceeded this limit. Most of the conclusions of this analysis remained the

same even if the limit was increased to $1000 \ \mathrm{cm}^{-3}$ or decreased to $100 \ \mathrm{cm}^{-3}$. Out of all the data in clean marine air, 12.7 % had N1-10 over $500 \ \mathrm{cm}^{-3}$. This is close to the fraction of time during which we observed growth starting from nucleation mode. With this data partition, we can see that global radiation levels are similar independent of N1-10 levels (Fig. 13a). This is not surprising, since our previous results showed that N1-10 could be high even during the night. Marine cluster formation cannot thus be explained by photochemistry alone.

With temperature, we can see at that times when N1-10 is higher, temperatures are on average lower (Fig. 13b). This is logical, because low temperatures can favour NPF by increasing nucleation rates (see for example, Burkholder et al. (2004); Simon et al. (2020)). For relative humidity (Fig. 13c), we see lower values when N1-10 is high. This is reasonable since high relative humidity can be related to weather with fog or low rainy clouds which would increase particle losses.

With wind speed, we can see higher N1-10, when wind speeds are lower (Fig. 13d). This makes sense because at high

wind speeds waves and sea spray aerosol production would typically be higher and sea spray aerosol can act as a sink for the smallest particles and their chemical precursor species. This can be also seen with the condensation sink, which is on average





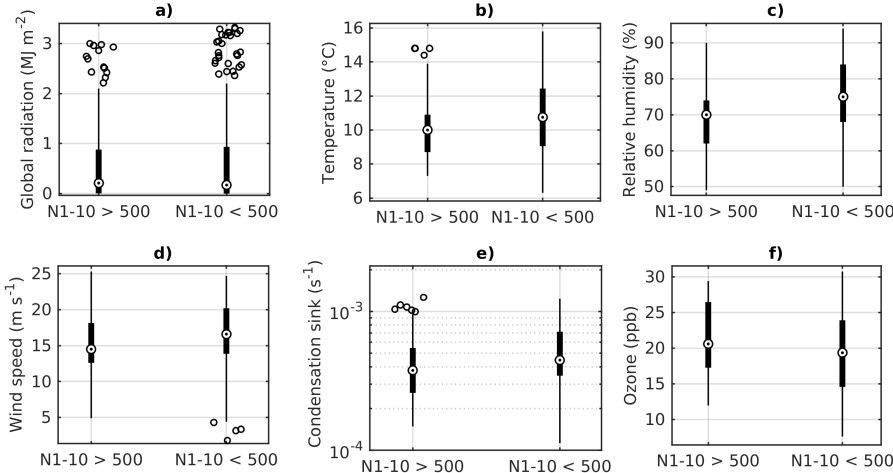

**Figure 13.** Box plots for different meteorological variables in marine air when N1-10 is a) over 500 b) and under 500.

lower when N1-10 is high (Fig. 13e). Previous work at Baring Head has estimated that the CS at the station would be too high for NPF to occur (Cainey and Harvey, 2002). Their work focused only on particle formation from $SO_2$ from marine sources, but since then, many other particle precursors have been identified, meaning that even if nucleation from $SO_2$ was unlikely,
nucleation from other precursors can still occur.

With ozone, we see slightly higher values when N1-10 is high (Fig. 13f), but the differences are small. Ozone could play some role in the chemical processes that influence particle formation in marine air, but the exact mechanisms cannot be studied with this data.

This analysis shows that initial particle formation in marine air is favoured by low temperatures, low relative humidity, and
low wind speeds. While this is in line with what we saw with the traditional event analysis, the traditional events were also favoured by high global radiation levels and likely driven by photochemistry, which is not true for particle formation in marine air. Our future research will study the different chemical species observed at Baring Head and look deeper into the factors controlling NPF at the site.

## 4   Conclusions

We studied new particle formation and typical aerosol number concentrations at Baring Head, New Zealand. The site is remote and enables studying clean marine air masses. During our 10 month measurement period, the average event frequency was 23 %, with the least events observed during the winter. These events detected with a traditional method designed for continental sites occurred primarily in land-influenced air and were favoured by high global radiation levels, low relative humidity, and low wind speeds. Aerosol number concentrations in all size ranges were significantly higher in land-influence air compared to
clean marine air. The concentrations increased when the air mass spent more time over land, with accumulation mode particle

concentrations doubling in a day, showing that aerosol production over New Zealand could have an effect on the regional cloud formation and properties.

In clean marine air, clear new particle formation events, when detected according to the NPF classification methods made for continental sites, were rare. However, we observed both sub-3 nm particles and particle growth starting from the nucleation mode in air masses that had only spent time within the marine boundary layer, showing that nucleation can happen within the marine boundary layer. Whilst these events do occur, they are weaker than terrestrially influenced NPF events. Unlike at some other coastal sites, coastal sources did not seem to play a significant role in aerosol formation at Baring Head. Formation of sub-10 nm particles was favoured by low temperatures, relative humidity and wind speeds. Our results highlight the need to study marine NPF with different criteria than continental NPF.

During our measurements, only 7.3 % of the data could be classified as clean marine air. In the future, it would be good to continue the measurements over longer periods to obtain more information on the importance of new particle formation in open ocean air. Our future work will focus on identifying the chemical precursors of new particle formation and growth at the site to provide a more complete picture of factors driving the particle concentrations both at the site in general and specifically in open ocean air.

*Data availability.* The meteorological data can be downloaded from https://cliflo.niwa.co.nz/ and tide height data from https://www.linz.govt.nz/sea/tides/tide-predictions. Wave height data is available from Greater Wellington Regional Council (http://graphs.gw.govt.nz/). The aerosol data will be made available on AERIS data base before final publication.

*Author contributions.* MP, JT and KS performed the aerosol measurements. MP and CR analysed the data. SG processed the radon data. MP wrote the paper with contributions from all authors. KS, CR and MH supervised the work.

*Competing interests.* The authors declare that they have no conflict of interest.

*Acknowledgements.* These results are part of a project that has received funding from the European Research Council (ERC) under the European Union's Horizon 2020 research and innovation programme (Grant agreement No. 771369). The Sea2Cloud project is endorsed by SOLAS. The authors gratefully acknowledge the NOAA Air Resources Laboratory (ARL) for the provision of the HYSPLIT transport and dispersion model and READY website (https://www.ready.noaa.gov) used in this publication. Supporting measurements from the Baring Head site are funded through NZ MBIE Strategic Science Investment Fund programme "Understanding Atmospheric Composition and Change". Sylvia Nichol is acknowledged for processing the ozone data.



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



## Appendix A: Tables

**Table A1.** Percentage of data available for each instrument each month calculated based on 30 minute averaged data.

| Month | PSM A09 | PSM A11 | CPC3010 | CPC3760A | SMPS | NAIS |
|---|---|---|---|---|---|---|
| April 2018 | 0 | 0 | 0 | 0 | 33 | 0 |
| May 2018 | 0 | 0 | 0 | 0 | 96 | 0 |
| June 2018 | 0 | 0 | 0 | 0 | 38 | 0 |
| June 2020 | 2 | 0 | 0 | 0 | 46 | 0 |
| July 2020 | 11 | 0 | 31 | 0 | 94 | 0 |
| August 2020 | 63 | 0 | 68 | 0 | 78 | 81 |
| September 2020 | 19 | 38 | 97 | 0 | 88 | 100 |
| October2020 | 0 | 97 | 95 | 0 | 82 | 100 |
| November 2020 | 0 | 86 | 86 | 0 | 76 | 100 |
| December 2020 | 0 | 51 | 53 | 24 | 73 | 100 |
| January 2021 | 0 | 62 | 0 | 94 | 64 | 97 |
| February 2021 | 0 | 46 | 0 | 59 | 94 | 59 |

## Appendix B: Extra figures





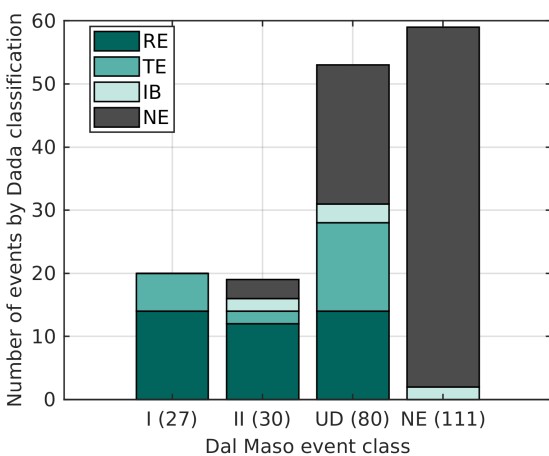

**Figure A1.** Comparison of the results of two event classification methods. The number in parenthesis shows the number of events in each class by the Dal Maso et al. classification. RE stands for regional events, TE for transported events, IB for ion bursts and NE for non-events.

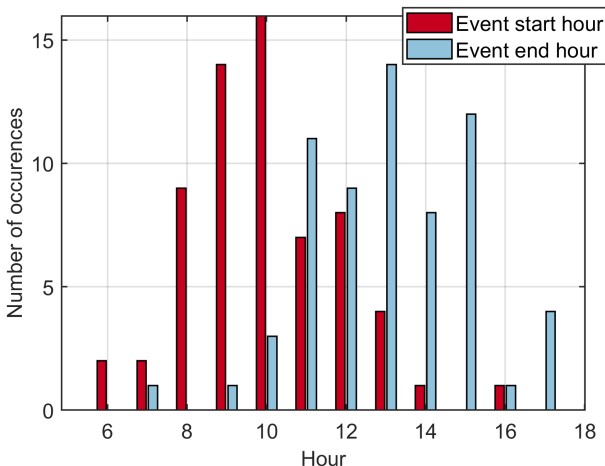

**Figure A2.** Bar plot shows the occurrence of event start and end times defined by the Dada et al. method during each hour of the day.





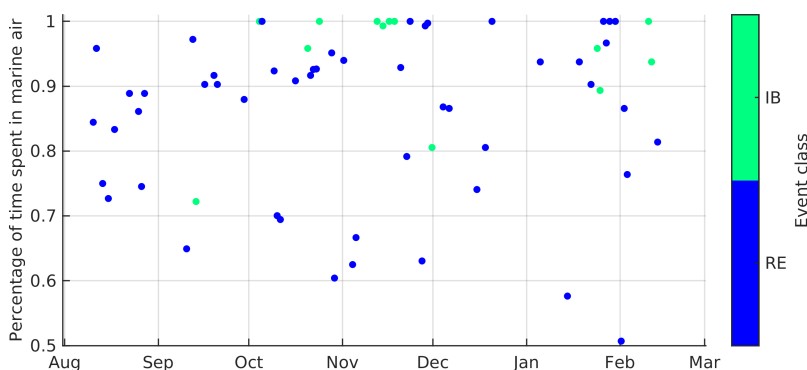

**Figure A3.** Percentage of time that back trajectories have spent in marine air during events coloured by the event class, where RE stands for regional events and IB for ion bursts.

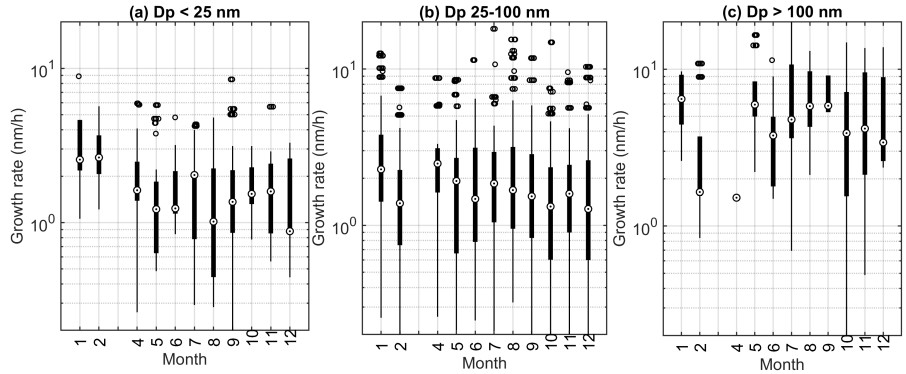

**Figure A4.** Seasonal cycles of growth rates in different modes.



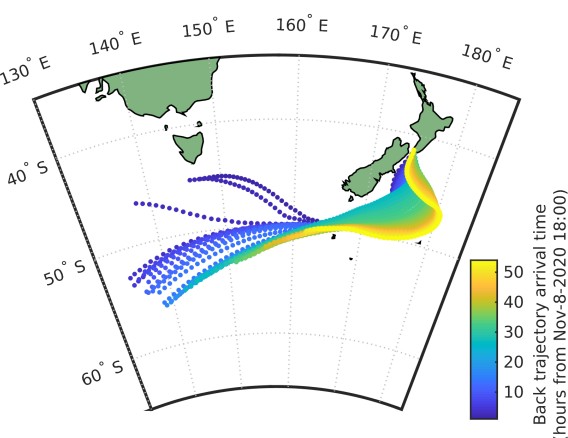

**Figure A5.** 72h air mass back trajectories for time from November 8th at 18:00 to end of November 10th 2020 coloured by arrival time at the station.