# Peer review of "New particle formation in coastal New Zealand with a focus on open ocean air masses"

_Atmospheric Chemistry and Physics, 2021_

## Referee Comment (RC1)

**1 General comments**

This is an interesting dataset and a data analysis that I feel falls well within the scope of ACP. The work illuminates the processes governing the aerosol distribution over the southern ocean as well as the impact of land masses on the aerosol size distribution, complementing and extending observations made elsewhere. I found the paper itself somewhat confusing, and suggest some refocussing or rewording would help to get the conclusions more clearly understood. These changes are listed below.

In the introduction (L70) New Zealand is claimed to be an optimal place for studying marine air, and yet the dataset clearly shows this is not true for Baring Head. I suspect that there are other places within New Zealand that are much better places for studying the southern marine atmosphere although they may not be practical logistically. (The logistical component should not be down-played - these are not trivial measurements!) There are also a number of other sites in the southern ocean that could be less influenced by land, including Macquarie Island. As noted in line 328 only 7% of the measurements made were classified as clean marine, and in line 109 it is noted that for $CO_2$ less than 10% of the time is classified as "baseline" (which is also nominally clean marine), noting that this estimate is based on a much longer time series of measurements. This "optimal" claim therefore colours the study of the general aerosol observations, which are overwhelmingly land influenced. This is unfortunate as that analysis is worthwhile.

I note that the same section (L 69) states that New Zealand is far from major pollution sources (meaning those external to New Zealand, although this is not explicit). In the context of aerosol lifetimes (say 1 day or less??) this is generally true although there has been significant aerosol deposits detected in New Zealand from dust and bushfire events in Australia. The statement assumes that there are no significant aerosol sources within New Zealand, which is not true in general and the impact on a measurement set is going to depend critically on site. Given that Baring head is less than 10 km from Wellington it is not really remote.

The second area where I struggled with this paper is in the size definitions used. The terms coarse mode,, climate relevant sizes, nucleation mode, Aitken mode, accumulation mode, N1-10, N10-100 (and N10) and N100 are all used and not well related. This is problematic as there is not a single definition for some of these terms, most critically for Aitken mode (10 - 25 nm lower limit, 80 - 100 nm upper limit) but also for nucleation mode (where in general the upper limit of 10 nm is agreed but the lower limit is often defined by instrumental limitations). A paragraph which defines the limits you are going to use for your work (and relate that to the measurements you report would significantly improve the clarity. I note that this problem is not unique to this work!

Definition of NPF events should be discussed more generally - given that you are then going to say means that it doesn't cut it for this dataset. Also discuss the implications of the definition of a NPF event - what is required of the meteorology for the definition to make sense. (In my opinion this is often left

unstated and it is important to realise the meteorological changes - in essence a change in the source function - can readily be interpreted as particle growth. Also need a shout out about noisy data.

**2   Specific Comments**

Aerosol measurements and inlet losses: it is difficult to assess as a reader how large the likely losses would be in the inlets for the various instruments. It would be beneficial if some estimate could be given for the losses for the various instruments, or at least an estimate of the difference in losses for the instruments, at different aerosol sizes.

**2.1   Significant figures**

Given the size of the dataset and the difficulty of clearly distinguishing the various processes that drive aerosol size distributions, the number of significant figures used seems misleading. This is doubly true as the total number of measurements relevant to these estimates is unclear. It is notable that the number of significant figures used in the abstract is less than that used in the general text, suggesting that the authors are somewhat aware of this. I suggest that, for example, L261 "10.9% of the days" is replaced with x out of y days (11%). Alternatively, if this becomes cumbersome there could be a table of the number of measurements that fall in each category. Further, at the same location in the text, the days included Class 1 events - the day itself was not a Class 1 event. This confuses the story as well. So on L261 it is clear that 10.9% is days/days, but the next sentence says "12.1% of the data" which is presumably days/ days, not measurements/measurements as the text would imply. I note that the information provided in Figure 1 is insufficient for the reader to assess the number of measurements or days used in the analyses.

**2.2   Section 3.1.2 - Factors favourig npf event occurance**

This section left me uncertain, as the there are likely to be a range of correlated variables that would confound an assignment of causation. If land masses are the primary source of npf events, you could expect a correlation of npf events with warmer conditions, clearer skies, etc.Indeed, if you look at Figure 4 it would seem that a label of "Land" for the Event category and "Ocean" for the Non-event category would make great sense of the observations, including the observed ozone concentrations (air impacted by NOx will have lower ozone concentrations). I suggest making this point clearer. It may be necessary to restrict this comparison to those days with significant land contact to make any firm conclusions.

**2.3 Land influence**

Figure 7 and the associated commentary. The text suggests that there is a clear relationship between time over land and particle concentration. Given the plots and the correlation coefficients quoted (significant??) this seems unconvincing. I think that this partially because of the way the data are presented. I suggest binning the data (by hour ranges) and showing the median (and I suggest the uncertainty of the medians) and the 75/95% points for the bins. These may more clearly show what you are trying to infer.

**2.4 Coastal effects**

L491: " coastal sources are likely not important for particle formation at Baring Head". The text then goes on to say this is really based on Mace Head observations, which is a very different location and may not apply here. It should be noted that there is some evidence of coastal sources of atmospheric iodine and new particle formation events at Cape Grim which also did not correlate with tide.(Grose et al., 2007) The studies of npf near the Great Barrier Reef also showed no real tidal signal but they considered that the source was most likely coastal. (Modini et al., 2009) This section needs a clearer message and conclusion and not rely so heavily on tide data for that conlusion.

**2.5 Cloud Processing of aerosol distribution**

This topic area which needs more consideration can most readily be seen in figure 11. The bi-modal structure appears to indicate that there is significant cloud processing of the aerosol for the entire period (as noted by the Hoppel minimum comment on L565. How does this change the size distributions as cloud processing varies, especially given the time scale of cloud processing versus the timescale used for considering npf? How can you distinguish the changes caused by cloud processing from gas phase particle growth?

L572 - "One key message from our work is that marine secondary aerosol formation should not be studied with the same criteria as continental new particle formation." This paper would be much stronger if this point was accompanied by some suggestion of what criteria should be used. Are you suggesting that looking at 2 - 4nm particle concentrations are more appropriate? I sort of think so.

**3 Technical Corrections**

L76, L109 - correct subscript
L171 - correct superscript.
L182 - 2 - 4 nm diameter.
L205 How many days' of data?
L277 - "Finokalia which has a more similar distance to the equator." Why is the distance to the equator important?

L297 - "the start height of the free troposphere can be variable" really means the height of the planetary boundary layer is variable.

L318 "saw nighttime increases in sub-3 and sub-10 nm particle concentrations" - occasionally? Often? Some indication of prevalence would be useful.

L320 - "To see more quantitatively" - To assess

L326 - radon concentration below

L327 "recently passed the southern tip of North Island" - given that the measurements are near the southern tip of the North Island it seems almost a give. If radon is your indicator presumably this means recent is in the last day?

L350 - "and we saw earlier" - correct the English

L352 - "With Condensation sink" ? Do you mean "Considering the variation in the calculated condensation sink"?

L352 - "slightly higher" is it significant?

L357 - Sentence starting "Over land" does not make sense.

L371 - reference the section where these limits are defined (2.3.1??)

L375 - "typically cleaner" has a very broad meaning. You mean that such differences are expected given the predominance of intense aerosol sources on land.

L376 - delete previous

L381 - replace numbers with observations (or aerosol concentrations)

L387 - not clear how the frequency of observations could create a trend, unless you are talking about the uncertainty in the medians??

L397 "A similar cycle"

L399 - is there a seasonal cycle in rainfall at this location?

L408 Figure 5 The end of the boxes are hard to detect (black box, black points). Suggest make box hollow or wider so that it is easier to distinguish.

L411 - the cycle depends on both the production and loss terms, and you should mention your assumption on the diurnal cycle of the loss term. Is this the cause of the early evening concentration increase??

L412 "rush hour" - the fact that you do not see a signal relevant to traffic probably should be clearer. Given the close proximity to Wellington it is an open question as to whether the impact of the city can be seen in the record.

L418 "time over land" - this needs definition in terms of the time window considered (in the last 72 hours?) and any vertical-based filter.

L439 - make it clear that the first part of your assessment is for all data.

L478 - "of ours" it was not measurements of you - it is "measurements reported here" or "our measurements at Baring head"

L528 - It appears that figure 10 shows patterns dominated by a small number of high particle concentration events - events that may have nothing to do with the more distant part of the track. Is there enough data here for these patterns to be interpretable? For example, the cluster "near Tasmania"- is it due to emissions from New Zealand rather than Tasmania? It is unclear to me.

L550 - y axis label in Figure 11 is wrong. Marking the start of the days with dates (9th/10th) would make it much easier to follow in the text. Given that Figure 12 supports Figure 11 it would be much better if the plotted area

aligned in the two figures - which is probably best achieved by combining the two plots.

**References**

Grose, M. R., Cainey, J. M., McMinn, A., Gibson, J. A. E., Grose, M. R., Cainey, J. M., ... Gibson, J. A. E. (2007). Coastal marine methyl iodide source and links to new particle formation at Cape Grim during February 2006. *Environmental Chemistry*, *4*(3), 172–177. doi:10.1071/EN07008

Modini, R. L., Ristovski, Z. D., Johnson, G. R., He, C., Surawski, N., Morawska, L., ... Kulmala, M. (2009). New particle formation and growth at a remote, sub-tropical coastal location. *Atmospheric Chemistry and Physics*, *9*(19), 7607–7621.

---

## Author Response (AR1)

*We thank the reviewers for their helpful feedback. Our replies to their comments are in red and italics.*

**Reviewer 1**

**1 General comments**
This is an interesting dataset and a data analysis that I feel falls well within the scope of ACP. The work illuminates the processes governing the aerosol distribution over the southern ocean as well as the impact of land masses on the aerosol size distribution, complementing and extending observations made elsewhere. I found the paper itself somewhat confusing, and suggest some refocussing or rewording would help to get the conclusions more clearly understood.

These changes are listed below.

In the introduction (L70) New Zealand is claimed to be an optimal place for studying marine air, and yet the dataset clearly shows this is not true for Baring Head. I suspect that there are other places within New Zealand that are much better places for studying the southern marine atmosphere although they may not be practical logistically. (The logistical component should not be down-played - these are not trivial measurements!) There are also a number of other sites in the southern ocean that could be less influenced by land, including Macquarie Island. As noted in line 328 only 7% of the measurements made were classified as clean marine, and in line 109 it is noted that for CO2 less than 10% of the time is classified as "baseline" (which is also nominally clean marine), noting that this estimate is based on a much longer time series of measurements. This "optimal" claim therefore colours the study of the general aerosol observations, which are overwhelmingly land influenced. This is unfortunate as that analysis is worthwhile.

*The reviewer is correct that logistical components play a major role in the choice of a station as well. Baring Head is a well established station accessible by car and it has been successfully used for baseline atmospheric measurements already for ~50 years. Conditions under which non contaminated marine air is received are well characterised for the station and combine use of radon, high resolution meteorological modelling, CO2 variability threshold and MSL pressure difference (east coast to west coast). We do thus think that it is a good place for long term aerosol measurements as well, even though only a fraction of the data can be classified as clean marine air.  We changed the word 'optimal' to 'compelling' on line 69 and 'beneficial' on line 70. The text now reads:*
*"It is thus a compelling place for studying marine air. Despite the beneficial location, –"*

I note that the same section (L 69) states that New Zealand is far from major pollution sources (meaning those external to New Zealand, although this is not explicit). In the context of aerosol lifetimes (say 1 day or less??) this is generally true although there has been significant aerosol deposits detected in New Zealand from dust and bushfire events in Australia. The statement assumes that there are no significant aerosol sources within New Zealand, which is not

true in general and the impact on a measurement set is going to depend critically on site. Given that Baring head is less than 10 km from Wellington it is not really remote.

*The reviewer is correct that the text could be more explicit. Even though Wellington city is only 10 km from the station and aerosol transport from Australia is possible, the station is on the coast and in suitable weather conditions, the air masses arriving at the station have not been in touch with land for several days nor affected by pollution from Wellington. During these times the sampled air is very clean and it corresponds well to data measured in remote conditions as discussed in Section 3.2.*

*To take into account aerosol sources on land around New Zealand, we changed the text so that the pollution sources on the island are mentioned and it now reads: "New Zealand is a contrasting environment as it is in the Southern Hemisphere, in the middle of the ocean, far from major pollution sources other than the sources on the islands themselves." Concerning pollution from Australia, Figure 10a shows that it is relatively rare that the air masses arriving at the station would have crossed over Australia during the past 72 h. This further illustrates that the influence of pollution from Australia is not likely a major factor at the station.*

The second area where I struggled with this paper is in the size definitions used. The terms coarse mode„ climate relevant sizes, nucleation mode, Aitken mode, accumulation mode, N1-10, N10-100 (and N10) and N100 are all used and not well related. This is problematic as there is not a single definition for some of these terms, most critically for Aitken mode (10 - 25 nm lower limit, 80 - 100 nm upper limit) but also for nucleation mode (where in general the upper limit of 10 nm is agreed but the lower limit is often defined by instrumental limitations). A paragraph which defines the limits you are going to use for your work (and relate that to the measurements you report would significantly improve the clarity. I note that this problem is not unique to this work!

*It is true that these size limits can seem confusing. We added some text to clarify the choices.*
*Text added on line 190:*
*"N1-10 represents the smallest particles that were measurable here and it is suitable for tracking particle formation, N10-100 contain both nucleation and Aitken mode particles and N100 is for accumulation mode particles."*
*Text added to line 441:*
*"To stay in line with the work by Paasonen et al. (2018), here the limits for nucleation, Aitken and accumulation modes are >25 nm, 25-100 nm and > 100 nm, respectively."*

Definition of NPF events should be discussed more generally - given that you are then going to say means that it doesn't cut it for this dataset. Also discuss the implications of the definition of a NPF event - what is required of the meteorology for the definition to make sense. (In my opinion this is often left unstated and it is important to realise the meteorological changes - in essence a change in the source function - can readily be interpreted as particle growth. Also need a shout out about noisy data.

*As Kulmala et al. (2012) state, NPF consists of the initial formation of particles and their growth to larger sizes. Whereas more traditional NPF classifications focus on regional NPF events where the air masses are uniform for several hours and the event can be tracked several hours from its start (see e.g. Dada et al. 2018), NPF can occur also over shorter periods or the measured air masses can fluctuate so that the event cannot be followed. It is true that meteorological changes such as how long the air mass has spent over land and whether it has temporarily crossed strong particle precursor sources such as coastal or anthropogenic sources can influence the interpretation of the data. The growth that we observed in this work is relatively slow, so we assume that it is in principle not caused by such factors.*

*The growth periods are also only defined if the growth is uniform enough for several hours. This should get rid of some of the effect of changes in meteorological conditions since it is unlikely that the conditions (air mass direction etc.) would change so that the particles grow uniformly for long enough.  The method by Paasonen et al. (2018) also smooths the data before looking for growth periods to get rid of small fluctuation in the mode diameter.*

*To clarify the NPF definition we added text to line 27:*
*" NPF refers to the formation of new clusters from gases and their subsequent growth to larger sizes (Kulmala et al., 2012)."*

*We also edited Section 2.3.3. To clarify what regional NPF events mean and what they require. We replaced "new particle formation is clear" on line 208 with "a clear regional NPF event can be observed" and added to line 209 "Class I and Class II events both require observing a particle mode that starts from below 25 nm and grows for several hours. This means that during the past hours before arriving at the station, the air masses have to cross a relatively uniform environment (in case of Dal Maso et al. (2005) a boreal forest spanning over 100 km in each direction from the station)."*

*We also modified the Results section to clarify how we interpret marine NPF.*
*Text added to line 489:*
*", meaning that both freshly formed particles and their later stage growth could be observed in marine air even if the phenomena were not strong or uniform enough to be classified as traditional NPF events"*

*Text added after line 573:*
*"In marine air, the sources of particle forming vapours can be weaker than over land, meaning that the events can have smaller formation and growth rates which means that they are more difficult to detect than typical regional events over land. At sites like Baring Head, where the prevailing winds and air masses can vary rapidly, it can be even harder to follow the events, since they might not fill the criteria that have been made for regional events that occur in more uniform air masses. The traditional methods are also often focusing on daytime data whereas we saw increased concentrations of sub-10 nm particles also during the night. We suggest measuring the particle size distribution down to sub-10 nm or even sub-3 nm sizes to detect the initial steps of particle formation and looking for trends in this data. One possibility is to separate between the initial formation and growth periods like we*

*have done in this study and investigate the factors influencing cluster formation and growth separately."*

**2 Specific Comments**

Aerosol measurements and inlet losses: it is difficult to assess as a reader how large the likely losses would be in the inlets for the various instruments. It would be beneficial if some estimate could be given for the losses for the various instruments, or at least an estimate of the difference in losses for the instruments,

*The figure below illustrates the diffusion losses in the lines of different aerosol instruments to give the reviewer a better image of the losses. Because the losses are so sensitive to particle size and thus uncertain when the measured size range is large (even difference between 1 nm and 2 nm can be substantial), we did not correct the data for the losses, but we added the following text to line 132 give a better image of the losses to the reader: "This is important because the smallest particles are the most sensitive to diffusion losses. With this inlet setup, the diffusion losses of 1 nm particles in the PSM inlet were 80 % for 1 nm particles and 6 % for 10 nm particles. For CPC and SMPS, the inlet diffusion losses of 10 nm particles were estimated to be 18 % and 21 %, respectively, while for 100 nm particles the losses are only 1 % for both instruments."*

[Figure]

**2.1 Significant figures**

Given the size of the dataset and the difficulty of clearly distinguishing the various processes that drive aerosol size distributions, the number of significant figures used seems misleading. This is doubly true as the total number of measurements relevant to these estimates is unclear. It is notable that the number of significant figures used in the abstract is less than that used in the general text, suggesting that the authors are somewhat aware of this. I suggest that, for example, L261 "10.9% of the days" is replaced with x out of y days (11%). Alternatively, if this becomes cumbersome there could be a table of the number of measurements that fall in each category. Further, at the same location in

the text, the days included Class 1 events - the day itself was not a Class 1 event. This confuses the story as well. So on L261 it is clear that 10.9% is days/days, but the next sentence says "12.1% of the data" which is presumably days/ days, not measurements/measurements as the text would imply. I note that the information provided in Figure 1 is insufficient for the reader to assess the number of measurements or days used in the analyses.

*The number of days used in this analysis is 258 (now mentioned in the methods; see reply to other reviewer). Table A1 shows the percentage of data available each month for different instruments based on 30 min resolution, so it gives a more precise image than Figure 1. We hope that this clears out the amount of available measurements used in this analysis. The amount of used days (over 100 days) means that reducing the number of significant figures would change the results if trying to move from percentage to days. For example 10% would round up to 26 days whereas 10.4% would round up to 27 days. Hence we would prefer not changing the text here. We have replaced "data" on line 261 by "days". The traditional event classification is always done day by day.*

**2.2 Section 3.1.2 - Factors favourig npf event occurance**

This section left me uncertain, as the there are likely to be a range of correlated variables that would confound an assignment of causation. If land masses are the primary source of npf events, you could expect a correlation of npf events with warmer conditions, clearer skies, etc.Indeed, if you look at Figure 4 it would seem that a label of "Land" for the Event category and "Ocean" for the Non-event category would make great sense of the observations, including the observed ozone concentrations (air impacted by NOx will have lower ozone concentrations). I suggest making this point clearer. It may be necessary to restrict this comparison to those days with significant land contact to make any firm conclusions.

*It is true that the conditions favouring traditional events are more common in land influenced air masses. It can be however more complicated than the reviewer indicates. For example for ozone, the impact of $NO_X$ is not the only influencing factor. Previous work at Cape Grim (similar to Baring Head) has shown that ozone has a seasonal cycle with a winter maximum of ~30 ppb and summer minimum of ~15-20 ppb (Galbally et al., 2013) and dry deposition velocity of $O_3$ to the ocean is smaller than to land so during Southerly onshore winds there are higher concentrations than during Northerly offshore winds due to this physical sink difference. Additionally, considering that the clean marine air consisted of only 7.3% of our data while non-events accounted for 44.8% of the days, we feel that removing the clean marine air and focusing only on land-influenced air would not change the results we are showing here. Extensive comparison of the effect of land influence on different parameters is outside the scope of this work, since we are more interested in aerosols and NPF.*

**2.3 Land influence**

Figure 7 and the associated commentary. The text suggests that there is a clear relationship between time over land and particle concentration. Given the plots and the correlation coefficients quoted (significant??) this seems unconvincing.

I think that this partially because of the way the data are presented. I suggest binning the data (by hour ranges) and showing the median (and I suggest the uncertainty of the medians) and the 75/95% points for the bins. These may more clearly show what you are trying to infer.

*We added 5 hour bins to the figure (including median, uncertainty of median with 95% confidence interval, 25th and 75th percentiles and whiskers for 1.0 times the interquartile range) and removed the second order fit as suggested by the second revier. The bins confirm that the particle concentrations increase as a function of time spent over land during the first ~25 h spent over land in all the size bins.*

*We removed the following text starting at line 420:*
*"For N100, particle concentrations increase during the first hours spent over land, but eventually they stabilise, with changes after 40 h being small. For sub-100 nm particles, there is an optimum time spent over land that is most favourable to particle formation. We used a second order polynomial fit to describe this time evolution. While the fits are not very good, and we have $R^2$'s of 0.03, 0.04 and 0.006 for N1-10, N10-100 and N100, respectively and root mean square errors of 1960, 1660 and 620 cm$^{-3}$ for N1-10, N10-100 and N100, respectively, the fits describe the general behaviour of particle concentrations. Both N1-10 and N10-100 reach their maximum concentration after the air mass has spent around 37 h over the land, but N100 keeps increasing and according to the fit it would only start decreasing after 101 h."*
*and replaced it by:*
*"N1-10 and N10-100 increase during the first ~25 h and after this the trends are less clear. N100 increases even after this with the median particle concentration increasing from 85 cm$^{-3}$ during the first 5 h spent over land to 380 cm$^{-3}$ after 65-70 h over land."*

[Figure]

**2.4 Coastal effects**

L491: " coastal sources are likely not important for particle formation at Baring Head". The text then goes on to say this is really based on Mace Head observations, which is a very different location and may not apply here. It should be noted that there is some evidence of coastal sources of atmospheric iodine and new particle formation events at Cape Grim which also did not correlate with tide.(Grose et al., 2007) The studies of npf near the Great Barrier Reef also showed no real tidal signal but they considered that the source was most likely coastal. (Modini et al., 2009) This section needs a clearer message and conclusion and not rely so heavily on tide data for that conlusion.

*It is true that Mace Head is very different as a location and the Australian sites are more similar to Baring Head. We decided to use Mace Head as an example of coastal nucleation here since NPF has been studied there more extensively and conclusively. The paper by Modini et al. (2009) mentions that all of their NPF events were shaped like apples which is typical for coastal events. At Baring Head, apple type events are more rare, but they should be studied in more detail in the future to see if any signs of coastal sources can be found.*

*Figure 9 does also look into the effect of wind direction to see if more particles are formed when the air masses for example pass a coast. The figure shows no effect of wind direction on 2-4 nm ion concentrations. To highlight this more, we added text to line 499:*
*", which supports the hypothesis of not having significant coastal sources". We are currently working on a second paper which studies the chemical precursors of particle formation at Baring Head (including iodine oxides) and we will discuss this topic further there.*

*We added text "related to tide changes" to line 498 after "sources" to highlight that this is focused on tides.*

*To highlight the uncertainties of this analysis, we added text to line 504:*
*"As this analysis is rather simplified and tide height might not be a good indicator of coastal sources in places where tide variations are small (see e.g., Modini et al., 2009), potential effects of coastal sources could be studied in more detail in the future with a more sophisticated analysis. Our future work will focus on chemical precursors of NPF at Baring Head and this work can also shed more light on the importance of coastal sources at the site by exploring for example the behaviour of iodine oxides."*

**2.5 Cloud Processing of aerosol distribution**
This topic area which needs more consideration can most readily be seen in figure 11. The bi-modal structure appears to indicate that there is significant cloud processing of the aerosol for the entire period (as noted by the Hoppel minimum comment on L565. How does this change the size distributions as cloud processing varies, especially given the time scale of cloud processing versus the timescale used for considering npf? How can you distinguish the changes caused by cloud processing from gas phase particle growth?

*This is an interesting point and should be studied more in the future. One indicator of possible cloud processing is the appearance of the Hoppel minimum, but apart from that it is hard to distinguish between cloud processing and condensation of gas phase species with our data set. Comparison of growth rates in different size ranges could also give hints of different processes since observing higher growth rates in accumulation mode compared to Aitken mode indicates either particle (or cloud) phase processing of semivolatile species (see Paasonen et al. 2018). Since the focus of our study is NPF and not cloud processing, we would prefer not going into more details here.*

L572 - "One key message from our work is that marine secondary aerosol formation should not be studied with the same criteria as continental new particle formation." This paper would be much stronger if this point was accompanied by some suggestion of what criteria should be used. Are you suggesting

that looking at 2 - 4nm particle concentrations are more appropriate? I sort of think so.

*Studying NPF should be started from as small size range as possible. While the 2-4 nm ions have been observed to be a good indicator of NPF in a boreal forest and it is definitely a good variable to investigate, in our work, the sub-10 nm concentration turned out to work better in marine air. It should be noted that this could be also due to the issues that we had with the NAIS measuring the 2-4 nm ions. Nevertheless, we suggest extending the measurements to as small sizes as possible and potentially looking at the appearance of freshly nucleated particles and particle growth separately so that NPF processes can be observed even if the events are not regional.*

*We have added a suggestion after line 573:*
*"In marine air, the sources of particle forming vapours can be weaker than over land, meaning that the events can be weaker and more difficult to detect. At sites like Baring Head, where the prevailing winds and air masses can vary rapidly, it can be even harder to follow the events, since they might not fill the criteria that have been made for regional events that occur in more uniform air masses. The traditional methods are also often focusing on daytime data whereas we saw increased concentrations of sub-10 nm particles also during the night. We suggest measuring the particle size distribution down to sub-10 nm or even sub-3 nm sizes to detect the initial steps of particle formation and looking for trends in this data. One possibility is to separate between the initial formation and growth periods like we have done in this study and investigate the factors influencing cluster formation and growth separately."*

**3 Technical Corrections**
L76, L109 - correct subscript
*We have fixed this.*
L171 - correct superscript.
*We have fixed this.*
L182 - 2 - 4 nm diameter.
*We have fixed this.*
L205 How many days' of data?
*258 days, added in parenthesis in the text.*
L277 - "Finokalia which has a more similar distance to the equator." Why is the distance to the equator important?
*The distance to the equator tells about the available radiation and its seasonality.*
L297 - "the start height of the free troposphere can be variable" really means the height of the planetary boundary layer is variable.
*True, we changed this.*
L318 "saw nighttime increases in sub-3 and sub-10 nm particle concentrations" - occasionally? Often? Some indication of prevalence would be useful.
*This was not quantified, but the diurnal cycle of marine N1-10 (Figure 6d) shows that most of the highest hourly medians of N1-10 occur after 19 h with the highest 75th percentile observed at 23 h. This shows that high nighttime N1-10 was frequent. We added text " and Figure 6d" after "Section 3.5.3" on line 318.*
L320 - "To see more quantitatively" - To assess
*We have fixed this.*

L326 - radon concentration below

*We have fixed this.*

L327 "recently passed the southern tip of North Island" - given that the measurements are near the southern tip of the North Island it seems almost a give. If radon is your indicator presumably this means recent is in the last day?

*Yes, in the last day and probably the last hours. We removed "recently" and added " likely just before arriving at the station" after 'North Island'.*

L350 - "and we saw earlier" - correct the English

*We added 'as' between 'and' and 'we'.*

L352 - "With Condensation sink" ? Do you mean "Considering the variation in the calculated condensation sink"?

*We have fixed this and the start of the sentence now reads "When comparing the diurnal cycles of condensation sink (CS, Fig. 4e) on event and non-event days –".*

L352 - "slightly higher" is it significant?

*The difference is not significant, but here we focus on the general trends.*

L357 - Sentence starting "Over land" does not make sense.

*We have fixed this, the last 'land' was supposed to be 'sea'.*

L371 - reference the section where these limits are defined (2.3.1??)

*We have fixed it and added "as explained in Section 2.3.1" to the end of the sentence.*

L375 - "typically cleaner" has a very broad meaning. You mean that such differences are expected given the predominance of intense aerosol sources on Land.

*We replaced "typically cleaner" with "have typically lower aerosol concentrations".*

L376 - delete previous

*Deleted.*

L381 - replace numbers with observations (or aerosol concentrations)

*Replaced 'number' by 'concentrations'.*

L387 - not clear how the frequency of observations could create a trend, unless you are talking about the uncertainty in the medians??

*Yes, we are referring to the uncertainty. We added ", meaning that the results for marine data are more uncertain" to the end of the sentence.*

L397 "A similar cycle"

*We have fixed this.*

L399 - is there a seasonal cycle in rainfall at this location?

*The average monthly rainfall in Wellington is higher during the winter compared to the summer. The long term (1960 - 2019) median and mean annual rainfall is close to 1320 mm. Seasonality is not strong and there are moderate amounts of rainfall throughout the year. Typically the wettest months are winter (June/July ~130 mm), the driest months are in summer (Jan Feb ~80 mm, see figure below).  Data and official statistics are available via https://www.stats.govt.nz/indicators/rainfall  and with registration available to download from https://data.mfe.govt.nz/table/105055-rainfall-1960-2019/data/.*

*We added the following text at the end of sentence on line 399:*
*", since the average rainfall is higher during the winter (June-July mean ~130 mm) compared to the summer (January-February mean ~80 mm, see https://www.stats.govt.nz/indicators/rainfall, last accessed February 14th, 2022)."*

[Figure]

L408 Figure 5 The end of the boxes are hard to detect (black box, black points). Suggest make box hollow or wider so that it is easier to distinguish.

*We added whiskers marking the 1.5 times the interquartile range to the figure (see below). This separates the box of the 25th-75th range from the outlier points and makes the plot easier to read. We did the same for Figure 6, to keep these two figures in the same style and increase the readability of Figure 6 as well.*

[Figure]

*New version of Figure 5*

[Figure]

*New version of Figure 6*

L411 - the cycle depends on both the production and loss terms, and you should mention your assumption on the diurnal cycle of the loss term. Is this the cause of the early evening concentration increase??

*The reviewer is correct that particle losses are also important to take into account. For example variations in boundary layer height could change particle concentrations, but here we assume that the changes are primarily related to NPF because the daytime increase is so consistent in all size ranges. In the case of the early evening fluctuations of N1-10 pointed out by the reviewer, we assume that this has not been caused by changes in the loss terms, because if that was the case, we would see changes also in the larger size ranges.*

L412 "rush hour" - the fact that you do not see a signal relevant to traffic probably should be clearer. Given the close proximity to Wellington it is an open question as to whether the impact of the city can be seen in the record.

*As wind passes over the lower North Island during the prevailing northwesterly wind, distance from the NZ coast to the city is of order 10 km and the city is only a further ~10-15km upwind of the station so distances and transit times are relatively short. This means that even if the air masses passed over Wellington, the time they would have to interact with the emission sources of the city would be short and it is thus not surprising that we do not see any patterns indicating traffic sources.*

L418 "time over land" - this needs definition in terms of the time window considered (in the last 72 hours?) and any vertical-based filter.

*The maximum time considered was 72 h and there was no vertical limit. We added text " This analysis is based on the geographical location of the air masses during the past 72 h." to line 419.*

L439 - make it clear that the first part of your assessment is for all data.

*We added "in all the data without land-marine air division" to the end of the sentence.*

L478 - "of ours" it was not measurements of you - it is "measurements reported here" or "our measurements at Baring head"

*We replaced "ours" by "our results for Baring Head".*

L528 - It appears that figure 10 shows patterns dominated by a small number

of high particle concentration events - events that may have nothing to do with the more distant part of the track. Is there enough data here for these patterns to be interpretable? For example, the cluster "near Tasmania"- is it due to emissions from New Zealand rather than Tasmania? It is unclear to me.

*It is true that these plots are challenging to interpret and caution should be taken here. As mentioned on line 185, we have limited the data so that only grid points with at least 10 points (1 h resolution) are visible, and hence we believe that it is unlikely that the observed high concentration areas would be caused by a single event. While it cannot be said for certain, transport of aerosol precursor species from the direction of Tasmania looks likely to us. If the source was only New Zealand, we would expect to see a patch of higher concentrations over NZ rather than in distinct regions over the sea. We have added text to line 514 to highlight the uncertainties and advantages of this figure:*

*"This shows how it can be hard to separate between particles formed from a source close to the station such as Wellington city vs particles formed from a precursor source further away from the station such as Australia. Even though caution should be taken in any interpretations, the figure is still useful for identifying regions that favour NPF formation."*

L550 - y axis label in Figure 11 is wrong. Marking the start of the days with dates (9th/10th) would make it much easier to follow in the text. Given that Figure 12 supports Figure 11 it would be much better if the plotted area aligned in the two figures - which is probably best achieved by combining the two plots.

*We have fixed the units of the y label of Figure 11a and combined Figures 11 and 12 into one figure (see below). We feel that having the time of day on the x-axis is more relevant than the date since time of day can tell more about potential processes, so we did not change the x-axis ticks.*

[Figure]

**References**

Grose, M. R., Cainey, J. M., McMinn, A., Gibson, J. A. E., Grose, M. R., Cainey, J. M., . . . Gibson, J. A. E. (2007). Coastal marine methyl iodide source and links to new particle formation at Cape Grim during February 2006. Environmental Chemistry, 4 (3), 172–177. doi:10.1071/EN07008

Modini, R. L., Ristovski, Z. D., Johnson, G. R., He, C., Surawski, N., Morawska, L., . . . Kulmala, M. (2009). New particle formation and growth at a remote, sub-tropical coastal location. Atmospheric Chemistry and Physics, 9 (19), 7607–7621.

**Reviewer 2**

The manuscript presents analyses of comprehensive measurements of aerosol number size-distributions from 1 nm to 500 nm, together with supporting data on meteorological variables and airmass back-trajectories.

The dataset presented is valuable and the results provide information on atmospheric new particle formation (NPF) occurrence from the Southern hemisphere and at a marine airmasses influenced area. This represents an environment type from where there are only few previous similar analyses in the literature, as most atmospheric NPF studies are concentrated on Northern hemisphere continental areas. The manuscript is within the scope of Atmos. Chem. Phys. Besides the comments given by Referee #1, I have listed below my additional comments. After considering these comments, I would recommend the manuscript to be published.

**General comments**

I suggest adding to Section 2.1 a map, which shows the location of the measurement site in New Zealand, with major cities indicated. This would help the reader to have a clearer picture of how the site is situated with respect to sources of at least some anthropogenic emissions (which was also pointed out by Referee #1). The sector used for the marine air mass arrival direction (120-220°) could be indicated in this figure, since marine airmasses are one of the focus points of the study.

*Considering the length of the manuscript, we would prefer not adding a map here. The location of the station is shown in Figure 10 and previous work by e.g. Stephens et al. (2013) contains both a map of the location of the station and urban areas and a figure with statistics for typical air mass source regions. We refer the readers unfamiliar with the station to this paper. To highlight this, we replaced the sentence "The site is described by Stephens et al. (2013)." with "The site, surrounding land areas and typical air masses are described by Stephens et al. (2013) and the location of the station is also shown in Figure 10 of this article." (line 95-100).*

In Section 2.3.2 (page 7, lines192–194) it is stated that the median of the ratio of the concentrations of the CPC and SMPS was used to correct the concentrations of the SMPS. How much does this ratio of the concentrations vary? I would expect it to depend on which size particles the peak of the size-distribution is. Can you estimate, how much uncertainty it introduces to the calculated particle formation rates, when you use a single correction value to the whole SMPS data timeseries?

*The 25th and 75th percentiles of the ratio are 1.15 and 2.01 and we have added this to the text. It is true that the ratio could vary based on the particle size distribution since the line losses are more important for the smaller particles and the line losses are larger for the SMPS than for the CPC, but we estimate this effect to be small. We briefly checked how the relationship of $N_{tot,\ CPC}$ /$N_{tot,SMPS}$ varies as function of the concentration of 10-25 nm particles since this size range is more sensitive to line losses than larger particles, meaning that when these particles are abundant, the losses in the SMPS would increase more compared to the CPC. We did not observe a statistically significant correlation (R = 0.02, p = 0.19) and we thus assume that the variations in the size distribution do not play a significant role in the correction factor. While changes in particle concentrations are important for the formation rate calculation, we assume that this correction and the given uncertainties would not change the order of magnitude of the formation rates.*

Section 3.1.1 (page 10, lines 286–289) discusses the influence of time that airmasses spent over land on the NPF characteristics. In the text, you refer to Fig. 3 which however shows the fraction of time spent in marine air. This feels a bit difficult to understand, how do you see the results presented in the text from Fig 3. Are the marine and land-based airmasses complementary to each other (or can there be airmass which is not classfied to either of those categories)? I would suggest to present in Fig 3 results of NPF characteristics as function of the time-over-land, which would more clearly relate to the discussion in the text.

*The air mass classes are complementary to each other, meaning that when fraction is 1, the air mass has spent the last 72 h over the ocean and when it is 0, the air mass has spent the past 72 h over land. We have fixed the figure (below) so that it uses hours spent in marine air instead and corresponds better to the text.*

[Figure]

Page 11, lines 296–297: Please give in the text the altitude which was used as the threshold between boundary layer and free troposphere. HYSPLIT model output could also be used to estimate the boundary layer height at each location along the trajectory (based on the input meteorological data). This could be potentially provide more insight into the division of boundary layer vs. free tropospheric airmasses, but I do not require that this type of further analyses needs to be added here.

*The altitude limit is 500 m and it has been mentioned on line 178, but we added it here too in parentheses after the word 'threshold' on line 296. It is true that more accurate estimation is possible but it was outside the scope of this work.*

In Figure 4, please specify are the lines medians or averages of the data. I strongly recommend adding some measure for the variability of the data, for example an area showing the 25th and 75th percentile (if the lines shown are median values) or standard

deviations (in case of averages). Based on this, the statistical significance of the differences between conditions on NPF and non-NPF days should be shortly discussed in the text.

*The lines stand for medians. We have fixed the caption and added shaded areas for 25th and 75th percentiles (see new figure below). Even though the shaded areas overlap, the trends of the percentiles are in line with the trends of the medians and thus in the text we focus on the trends of the medians. We added the following text to line 337:*
*"We draw the diurnal cycles of the medians and 25th and 75th percentiles of different variables separately for event and non-event days (Fig. 4). Although the ranges of 25th-75th percentiles overlap for all the variables, the trends of the percentiles are similar to the trends of the medians and here we focus on the trends of the medians. "*

[Figure]

In Figure 6d, the diurnal variation of N1-10 seems to have a clear pattern which is clearly different from the diurnal pattern in land-influenced N1-10: in marine air there seems to be an early morning peak (around 05:00) and this could be actually linked to the example case shown in Figure 11 (high N1-3 and N1-10 concentrations after midnight on 9th Nov 2020). This pattern should be more clearly highlighted in the text, since the marine NPF observations are one the most interesting findings of this study.

*We added the following text on line 417 to highlight this:*
*"There are, however, some trends in N1-10 even in marine air. The concentration peaks in the morning around 5 h, then decreases towards the afternoon and increases again in the afternoon and during the night. We further investigate nighttime N1-10 with a case study in Section 3.5.3. Different chemical mechanisms could be responsible for particle formation at different times of the day and this needs to be studied further in the future."*

In Figure 7, the second order polynomial fits should not be used. In my opinion they do not represent the trends of the datasets (as is also said in the manuscript), and there is also no physical reasoning why the particle concentrations would follow a second order polynomial

of the time-over-land. It's also potentially misleading to interpret anything based on the extrapolation of polynomial fits, as is done from Fig. 7c on line 426 ("… according to the fit it would only start decreasing after 101 h"). I suggest binning the data based on time-over-land and analyzing the trends in the distribution (medians and 25th–75th percentile) of the binned data (similar to the suggestion from Referee #1).

*This is true. We replaced the second order fits with box plots for data binned for every 5 h spent over land (see response to the other reviewer).*

The discussion related to Figure 13 on the differences between conditions during the cases of high N1-10 (>500 cm-3) and low N1-10 (<500 cm-3) seems to overstate these differences. The differences in temperature, RH and wind speed are highlighted in the text (lines 599–600) but for example ozone is said to have small difference between these cases (line 596). Only looking at Fig. 13 I would not draw this kind of conclusion. Have you done any statistical test, which could tell if there are statisitically significant differences at any of the studied variables between these two cases? For example, two-sample t-test could be used here.

*The reviewer is correct that the text can be confusing here. To clarify the statistical differences, we added notches that show the 95 % confidence interval of the medians to the figure. If the notches of two box plots do not overlap, the medians are different at 95 % confidence level. Inspecting this, we could see that the medians are statistically significantly different for temperature, relative humidity and wind speed.*

*We added the following text to end of the sentence on line 577:*
*"and compare the medians and their confidence intervals for different variables during high and low N1-10"*

*We also added  the following text to end of line 592:*
*", although this difference is not statistically significant at the 95 % confidence interval"*

*On line 596, we replaced "the differences are small" by:*
*"the difference is not statistically significant at the 95 % confidence interval"*

*The following text was added to the caption:*
*"The circles are the median concentrations, notches (triangles) show the 95 % confidence interval of the medians, black boxes mark 25th and 75th percentiles, the whiskers cover approximately 99.3 % of the data with rest of the points being outside this range"*

[Figure]

**Minor comments and technical corrections**

Page 3, line 62: "… can contribute between 92–49 % …", I suggest writing this the other way around as "49–92%"

*We have fixed this.*

Page 4, line 93, "… marine masses …", should be "… marine air masses …"

*We have fixed this.*

Page 5, lines 134–135: Why is the measurement range of ions reported as mobilities (electrical mobility is the primary variable of the NAIS in both particle and ion mode, as well as in SMPS)? It's not logical to write some of these in diameters and some in mobilities; I suggest stating also the ion measurements in nanometer size range.

*This was done because this is how it has been reported in previous work.*

Page 6, lines 144–145: Reference format should be without the dates "Chambers et al., 2014"

*We have fixed this.*

Page 9, line 266: "pre-exiting", should be "pre-existing"

*We have fixed this.*

Page 13, line 357: The end of the sentence "… more intense than over land" should probably be "… more intense than over sea".

*We have fixed this.*

Page 19, line 508: I suggest repling "… particle emitting precursor …" with "… particle forming precursor …" (a precursor is related to formation, not emissions)

*We have fixed this.*

Page 20, line 520: The highest ion concentrations in Fig. 10b are in southwestern, not northwestern airmasses.

*We have fixed this.*

Page 24, Figure 13 caption text: Add the explanations for the circles, box widths etc. (similarly as in Figs. 5, 6 and 8).

*We have fixed this.*

***References:***

*Dada, L., Chellapermal, R., Buenrostro Mazon, S., Paasonen, P., Lampilahti, J., Manninen, H. E., Junninen, H., Petäjä, T., Kerminen, V.-M., and Kulmala, M.: Refined classification and characterization of atmospheric new-particle formation events using air ions, Atmospheric Chemistry and Physics, 18, 17 883–17 893, 2018.*

*Dal Maso, M., Kulmala, M., Riipinen, I., Wagner, R., Hussein, T., Aalto, P. P., and Lehtinen, K. E.: Formation and growth of fresh atmospheric aerosols: eight years of aerosol size distribution data from SMEAR II, Hyytiala, Finland, Boreal Environment Research, 10, 323, 2005.*

*Galbally, I. E., Schultz, M. G., Buchmann, B., Gilge, S., Guenther, F., Koide, H., Oltmans, S., Patrick, L., Scheel, H., Smit, H., et al.: Guidelines for continuous measurement of ozone in the troposphere, World Meteorological Organization, Geneva, 2013*

*Kulmala, M., Petäjä, T., Nieminen, T., Sipilä, M., Manninen, H. E., Lehtipalo, K., Dal Maso, M., Aalto, P. P., Junninen, H., Paasonen, P., et al.: Measurement of the nucleation of atmospheric aerosol particles, Nature protocols, 7, 1651, 2012.*

*Paasonen, P., Peltola, M., Kontkanen, J., Junninen, H., Kerminen, V.-M., and Kulmala, M.: Comprehensive analysis of particle growth rates from nucleation mode to cloud condensation nuclei in boreal forest, Atmospheric Chemistry and Physics, 18, 12 085–12 103, 2018.*

*Stephens, B., Brailsford, G., Gomez, A., Riedel, K., Fletcher, S. M., Nichol, S., and Manning, M.: Analysis of a 39-year continuous atmo-spheric CO2 record from Baring Head, New Zealand, Biogeosciences, 10, 2683, 2013.*